# Impact of Representation Learning in Linear Bandits

**Jiaqi Yang**
Tsinghua University
yangjq17@gmail.com

**Wei Hu**
Princeton University
huwei@cs.princeton.edu

**Jason D. Lee**
Princeton University
jasonlee@princeton.edu

**Simon S. Du**
University of Washington
ssdu@cs.washington.edu

## Abstract

We study how representation learning can improve the efficiency of bandit problems. We study the setting where we play $T$ linear bandits with dimension $d$ concurrently, and these $T$ bandit tasks share a common $k(\ll d)$ dimensional linear representation. For the finite-action setting, we present a new algorithm which achieves $\widetilde{O}(T\sqrt{kN} + \sqrt{dkNT})$ regret, where $N$ is the number of rounds we play for each bandit. When $T$ is sufficiently large, our algorithm significantly outperforms the naive algorithm (playing $T$ bandits independently) that achieves $\widetilde{O}(T\sqrt{dN})$ regret. We also provide an $\Omega(T\sqrt{kN} + \sqrt{dkNT})$ regret lower bound, showing that our algorithm is minimax-optimal up to poly-logarithmic factors. Furthermore, we extend our algorithm to the infinite-action setting and obtain a corresponding regret bound which demonstrates the benefit of representation learning in certain regimes. We also present experiments on synthetic and real-world data to illustrate our theoretical findings and demonstrate the effectiveness of our proposed algorithms.

## 1 Introduction

This paper investigates the benefit of using representation learning for sequential decision-making problems. Representation learning learns a joint low-dimensional embedding (feature extractor) from different but related tasks and then uses a simple function (often a linear one) on top of the embedding (Baxter, 2000; Caruana, 1997; Li et al., 2010) The mechanism behind is that since the tasks are related, we can extract the common information more efficiently than treating each task independently.

Empirically, representation learning has become a popular approach for improving sample efficiency across various machine learning tasks (Bengio et al., 2013). In particular, recently, representation learning has become increasingly more popular in sequential decision-making problems (Teh et al., 2017; Taylor & Stone, 2009; Lazaric & Restelli, 2011; Rusu et al., 2015; Liu et al., 2016; Parisotto et al., 2015; Higgins et al., 2017; Hessel et al., 2019; Arora et al., 2020; D'Eramo et al., 2020). For example, many sequential decision-making tasks share the same environment but have different reward functions. Thus a natural approach is to learn a succinct representation that describes the environment and then make decisions for different tasks on top of the learned representation.

While representation learning is already widely applied in sequential decision-making problems empirically, its theoretical foundation is still limited. One important problem remains open:

**When does representation learning *provably* improve efficiency of sequential decision-making problems?**

We take a step to characterize the benefit of representation learning in sequential decision-making problems. We tackle the above problem in the linear bandits setting, one of the most fundamental

and popular settings in sequential decision-making problems. This model is widely used in applications as such clinical treatment, manufacturing process, job scheduling, recommendation systems, etc (Dani et al., 2008; Chu et al., 2011). We study the multi-task version of linear bandits, which naturally models the scenario where one needs to deal with multiple different but closely related sequential decision-making problems concurrently.

We will mostly focus on the finite-action setting. Specifically, we have $T$ tasks, each of which is governed by an unknown linear coefficient $\boldsymbol{\theta}_t \in \mathbb{R}^d$. At the $n$-th round, for each task $t \in [T]$, the player chooses an action $a_{n,t}$ that belongs to a finite set, and receive a reward $r_{n,t}$ with expectation $\mathbb{E}\, r_{n,t} = \langle \boldsymbol{\theta}_t, \boldsymbol{x}_{n,t,a_{n,t}} \rangle$ where $\boldsymbol{x}_{n,t,a_{n,t}}$ represents the context of action $a_{n,t}$. For this problem, a straightforward approach is to treat each task independently, which leads to $\widetilde{O}(T\sqrt{dN})$[1] total regret. Can we do better?

Clearly, if the tasks are independent, then by the classical $\Omega(\sqrt{dN})$ per task lower bound for linear bandit, it is impossible to do better. We investigate how representation learning can help if the tasks are related. Our main assumption is the existence of an unknown *linear feature extractor* $\boldsymbol{B} \in \mathbb{R}^{d \times k}$ with $k \ll d$ and a set of linear coefficients $\{\boldsymbol{w}_t\}_{t=1}^T$ such that $\boldsymbol{\theta}_t = \boldsymbol{B}\boldsymbol{w}_t$. Under this assumption, the tasks are closely related as $\boldsymbol{B}$ is a *shared* linear feature extractor that maps the raw contexts $\boldsymbol{x}_{n,t,a} \in \mathbb{R}^d$ to a low-dimensional embedding $\boldsymbol{B}^\top \boldsymbol{x}_{n,t,a} \in \mathbb{R}^k$. In this paper, we focus on the regime where $k \ll d, N, T$. This regime is common in real-world problems, e.g., computer vision, where the input dimension is high, the number of data is large, many task are related, and there exists a low-dimension representation among these tasks that we can utilize. Problems with similar assumptions have been studied in the supervised learning setting (Ando & Zhang, 2005). However, to our knowledge, this formulation has not been studied in the bandit setting.

**Our Contributions**   We give the first rigorous characterization on the benefit of representation learning for multi-task linear bandits. Our contributions are summarized below.

- We design a new algorithm for the aforementioned problem. Theoretically, we show our algorithm incurs $\widetilde{O}(\sqrt{dkTN} + T\sqrt{kN})$ total regret in $N$ rounds for all $T$ tasks. Therefore, our algorithm outperforms the naive approach with $O(T\sqrt{dN})$ regret. To our knowledge, this is the first theoretical result demonstrating the benefit of representation learning for bandits problems.

- To complement our upper bound, we also provide an $\Omega(\sqrt{dkTN} + T\sqrt{kN})$ lower bound, showing our regret bound is tight up to polylogarithmic factors.

- We further design a new algorithm for the infinite-action setting, which has a regret $\widetilde{O}(d^{1.5}k\sqrt{TN} + kT\sqrt{N})$, which outperforms the naive approach with $O(Td\sqrt{N})$ regret in the regime where $T = \widetilde{\Omega}(dk^2)$.

- We provide simulations and an experiment on MNIST dataset to illustrate the effectiveness of our algorithms and the benefits of representation learning.

**Organization**   This paper is organized as follows. In Section 2, we discuss related work. In Section 3, we introduce necessary notation, formally set up our problem, and describe our assumptions. In Section 4, we present our main algorithm for the finite-action setting and its performance guarantee. In Section 5, we describe our algorithm and its theoretical guarantee for the infinite-action setting. In Section 6, we provide simulation studies and real-world experiments to validate the effectiveness of our approach. We conclude in Section 7 and defer all proofs to the Appendix.

## 2 RELATED WORK

Here we mainly focus on related theoretical results. We refer readers to Bengio et al. (2013) for empirical results of using representation learning.

For supervised learning, there is a long line of works on multi-task learning and representation learning with various assumptions (Baxter, 2000; Ando & Zhang, 2005; Ben-David & Schuller, 2003; Maurer, 2006; Cavallanti et al., 2010; Maurer et al., 2016; Du et al., 2020; Tripuraneni et al., 2020).

---

[1] $\widetilde{O}(\cdot)$ omits logarithmic factors.

All these results assumed the existence of a common representation shared among all tasks. However, this assumption alone is not sufficient. For example, Maurer et al. (2016) further assumed every task is i.i.d. drawn from an underlying distribution. Recently, Du et al. (2020) replaced the i.i.d. assumption with a deterministic assumption on the input distribution. Finally, it is worth mentioning that Tripuraneni et al. (2020) gave the method-of-moments estimator and built the confidence ball for the feature extractor, which inspired our algorithm for the infinite-action setting.

The benefit of representation learning has been studied in sequential decision-making problems, especially in reinforcement learning domains. D'Eramo et al. (2020) showed that representation learning can improve the rate of approximate value iteration algorithm. Arora et al. (2020) proved that representation learning can reduce the sample complexity of imitation learning. Both works require a probabilistic assumption similar to that in (Maurer et al., 2016) and the statistical rates are of similar forms as those in (Maurer et al., 2016).

We remark that representation learning is also closely connected to meta-learning (Schaul & Schmidhuber, 2010). Raghu et al. (2019) empirically suggested that the effectiveness of meta-learning is due to its ability to learn a useful representation. There is a line of works that analyzed the theoretical properties of meta-learning (Denevi et al., 2019; Finn et al., 2019; Khodak et al., 2019; Lee et al., 2019; Bertinetto et al., 2018). We also note that there are analyses for other representation learning schemes (Arora et al., 2019; McNamara & Balcan, 2017; Galanti et al., 2016; Alquier et al., 2016; Denevi et al., 2018).

Linear bandits (stochastic linear bandits / linearly parameterized bandits / contextual linear bandits) have been studied in recent years (Auer, 2002; Dani et al., 2008; Rusmevichientong & Tsitsiklis, 2010; Abbasi-Yadkori et al., 2011; Chu et al., 2011; Li et al., 2019a;b). The studies are divided into two branches according to whether the action set is finite or infinite. For the finite-action setting, $\widetilde{\Theta}(\sqrt{dN})$ has been shown to be the near-optimal regret bound (Chu et al., 2011; Li et al., 2019a), and for the infinite-action setting, $\widetilde{\Theta}(d\sqrt{N})$ regret bound has been shown to be near-optimal (Dani et al., 2008; Rusmevichientong & Tsitsiklis, 2010; Li et al., 2019b).

Some previous work studied the impact of low-rank structure in linear bandit. Lale et al. (2019) studied a setting where the context vectors share a low-rank structure. Specifically, in their setting, the context vectors consist of two parts, i.e. $\hat{x} = x + \psi$, so that $x$ is from a hidden low-rank subspace and $\psi$ is i.i.d. drawn from an isotropic distribution. Jun et al. (2019) and Lu et al. (2020) studied the bilinear bandits with low-rank structure. In their setting, the player chooses two actions $x, y$ and receives the stochastic reward with mean $x^\top \Theta y$, where $\Theta$ is an unknown low-rank bilinear form. The algorithms proposed in the aforementioned papers share some similarities with our Algorithm 2 for our infinite-action setting, in that both used Davis-Kahan theorem to recover and exploit the low-rank structure.

Some previous work proposed multi-task bandits with different settings. Deshmukh et al. (2017) proposed a setting under the contextual bandit framework. They assumed similarities among arms. Bastani et al. (2019) studied a setting where the coefficients of the tasks were drawn from a gaussian distribution fixed across tasks and proposed an algorithm based on Thompson sampling. Soare et al. (2018) proposed a setting where tasks were played one by one sequentially and the coefficients of the tasks were near in $\ell_2$ distance. In our setting, the tasks are played simultaneously and the coefficients share a common linear feature extractor.

## 3 PRELIMINARIES

**Notation.** We use bold lowercases for vectors and bold uppercases for matrices. For any positive integer $n$, we use $[n]$ to denote the set of integers $\{1, 2, \ldots, n\}$. For any vector $x$, we use $\|x\|$ to denote its $\ell_2$ norm. For a matrix $A$, we use $\|A\|$ to denote the 2-norm of $A$, $\|A\|_F$ to denote the Frobenius norm, and $\|A\|_{\max} = \max_{i,j} |A_{ij}|$ to denote the max-norm. For two expressions $\alpha, \beta > 0$, we denote $\alpha \lesssim \beta$ if there is a numerical constant $c > 0$ such that $\alpha \leq c\beta$. We denote $\alpha \gtrsim \beta$ if $\beta \lesssim \alpha$.

**Problem Setup.** Let $d$ be the ambient dimension and $k (\leq d)$ be the representation dimension. In total, we have $T$ tasks and we play each task concurrently for $N$ rounds. Each task $t \in [T]$ has an

unknown vector $\boldsymbol{\theta}_t \in \mathbb{R}^d$. At each round $n \in [N]$, the player chooses action $\boldsymbol{a}_{n,t} \in \mathcal{A}_{n,t}$ for each task $t \in [T]$ where $\mathcal{A}_{n,t}$ is the action set at round $n$ for the task $t$.

After the player commits to a batch of actions $\{\boldsymbol{a}_{n,t}\}_{t \in [T]}$, it receives a batch of rewards $\{r_{n,t}\}_{t \in [T]}$, where we assume $r_{n,t} = \langle \boldsymbol{a}_{n,t}, \boldsymbol{\theta}_t \rangle + \varepsilon_{n,t}$. Here we assume the noise $\varepsilon_{n,t}$ are independent 1-sub-Gaussian random variables, which is a standard assumption in the literature.

We use the total expected regret to measure the performance of our algorithm. When we have $N$ rounds and $T$ tasks, it is defined as $R^{N,T} = \sum_{n=1}^{N} \sum_{t=1}^{T} \max_{\boldsymbol{a} \in \mathcal{A}_{n,t}} \langle \boldsymbol{a}, \boldsymbol{\theta}_t \rangle - \langle \boldsymbol{a}_{n,t}, \boldsymbol{\theta}_t \rangle$. When the action set is finite, we assume that all $\mathcal{A}_{n,t}$ have the same size $K$, i.e. $|\mathcal{A}_{n,t}| \equiv K$. Furthermore, we write $\mathcal{A}_{n,t} = \{\boldsymbol{x}_{n,t,1}, \ldots, \boldsymbol{x}_{n,t,K}\}$. Besides, we interchangeably use the number $a_{n,t} \in [K]$ and the vector $\boldsymbol{a}_{n,t} = \boldsymbol{x}_{n,t,a_{n,t}} \in \mathbb{R}^d$ to refer to the same action.

**Assumptions.** Our main assumption is the existence of a common linear feature extractor.

**Assumption 1** (Common Feature Extractor). There exists a linear feature extractor $\boldsymbol{B} \in \mathbb{R}^{d \times k}$ and a set of linear coefficients $\{\boldsymbol{w}_t\}_{t=1}^{T}$ such that the expected reward of the $t$-th task at the $n$-th round satisfies $\mathbb{E}[r_{t,n}] = \langle \boldsymbol{w}_t, \boldsymbol{B}^\top \boldsymbol{x}_{n,t,a_{n,t}} \rangle$.

For simplicity, we let $\boldsymbol{W} = [\boldsymbol{w}_1, \ldots, \boldsymbol{w}_T]$. Assumption 1 implies that $\boldsymbol{\Theta} \triangleq [\boldsymbol{\theta}_1, \ldots, \boldsymbol{\theta}_T] = \boldsymbol{BW}$. Note this assumption is in a sense necessary to guarantee the effectiveness of representation learning because without it one cannot hope that representation learning helps.

In this paper, we mostly focus on the finite-action setting. We put the following assumption on action sets.

**Assumption 2.** Marginally, for every $n \in [N], t \in [T], a \in [K]$, the contexts satisfy $\boldsymbol{x}_{n,t,a} \sim \mathcal{N}(\boldsymbol{0}, \Sigma_t)$ such that $\lambda_{\max}(\Sigma_t) \leq O(1/d)$ and $\lambda_{\min}(\Sigma_t) \geq \Omega(1/d)$.

With this assumption, we have an unknown covariance matrix $\Sigma_t$ for each task. At each round, the actions of the $t$-th task are sampled from a Gaussian distribution with covariance $\Sigma_t$. This is a prototypical setting for theoretical development on linear bandits with finite actions (See e.g., Han et al. (2020)). At a population level, each one of the $K$ actions is equally good but being able to select different actions based on the realized contexts allows the player to gain more reward.

We will also study the infinite-action setting. We first state our assumption about the action sets.

**Assumption 3** (Ellipsoid Action Set). We assume $\mathcal{A}_{n,t} = \mathcal{A}_t = \{x^\top \boldsymbol{Q}_t^{-1} x \leq 1 : x \in \mathbb{R}^d\}$ is an ellipsoid with $\lambda_{\min}(\boldsymbol{Q}_t) \geq \lambda_0 = \Omega(1)$.

The first assumption states that each action set is an ellipsoid that covers all directions. This is a standard assumption, e.g., see Rusmevichientong & Tsitsiklis (2010).

In this setting, we will also need to put some additional assumptions on the underlying parameters $\boldsymbol{B}$ and $\boldsymbol{W}$.

**Assumption 4** (Diverse Source Tasks). We assume that $\lambda_{\min}(\frac{1}{T} \boldsymbol{W}\boldsymbol{W}^\top) \geq \frac{\nu}{k}$, where $\nu = \Omega(1)$.

This assumption roughly states that the underlying linear coefficients $\{\boldsymbol{w}_t\}_{t=1}^{T}$ equally spans all directions in $\mathbb{R}^k$. This is a common assumption in representation learning literature that enables us to learn the linear feature extractor (Du et al., 2020; Tripuraneni et al., 2020). For example, the assumption holds with high probability when $\boldsymbol{w}_i$ is uniformly chosen from the sphere $\mathbb{S}^{k-1}$.

**Assumption 5.** We assume $\|\boldsymbol{w}_t\| \geq \omega = \Omega(1)$.

This is a normalization assumption on the linear coefficients.

## 4 MAIN RESULTS FOR FINITE-ACTION SETTING

In this section focus on the finite-action setting. The pseudo-code is listed in Algorithm 1. Our algorithm uses a doubling schedule rule (Gao et al., 2019; Simchi-Levi & Xu, 2019; Han et al., 2020; Ruan et al., 2020). We only update our estimation of $\boldsymbol{\theta}$ after an epoch is finished, and we only use samples collected within the epoch. In Line 5, we solve an empirical $\ell_2$-risk minimization problem on the data collected in the last epoch to estimate the feature extractor $\boldsymbol{B}$ and linear predictors

---

**Algorithm 1:** MLinGreedy: Multi-task Linear Bandit with Finite Actions

---

1: Let $M = \lceil \log_2 \log_2 N \rceil, \mathcal{G}_0 = 0, \mathcal{G}_M = N, \mathcal{G}_m = N^{1-2^{-m}}$ for $1 \leq m \leq M-1$, let $\widehat{\boldsymbol{\theta}}_{0,t} \leftarrow \mathbf{0}$;

2: **for** $m \leftarrow 1, \ldots, M$ **do**

3:     **for** $n \leftarrow \mathcal{G}_{m-1} + 1, \ldots, \mathcal{G}_m$ **do**

4:         For each task $t \in [T]$: choose action $a_{n,t} = \arg\max_{a \in [K]} \boldsymbol{x}_{n,t,a}^\top \widehat{\boldsymbol{\theta}}_{m-1,t}$;

5:     Compute $\widehat{\boldsymbol{B}}\widehat{\boldsymbol{W}} \leftarrow \underset{\boldsymbol{B} \in \mathbb{R}^{d \times k}, \boldsymbol{W} \in \mathbb{R}^{k \times T}}{\arg\min} \sum_{n=\mathcal{G}_{m-1}+1}^{\mathcal{G}_m} \sum_{t=1}^{T} [\boldsymbol{x}_{n,t,a_{n,t}}^\top \boldsymbol{B}\boldsymbol{w}_t - r_{n,t}]^2$;

6:     For each task $t \in [T]$: let $\widehat{\boldsymbol{\theta}}_{m,t} = \widehat{\boldsymbol{B}}\widehat{\boldsymbol{w}}_t$;

---

$\boldsymbol{W}$, similar to Du et al. (2020). Given estimated feature extractor $\widehat{\boldsymbol{B}}$ and linear predictors $\widehat{\boldsymbol{W}}$, we compute our estimated linear coefficients of task $t$ as $\widehat{\boldsymbol{\theta}}_t \triangleq \widehat{\boldsymbol{B}}\widehat{\boldsymbol{w}}_t$ in Line 6. For choosing actions, for each task, we use a greedy rule, i.e., we choose the action that maximizes the inner product with our estimated $\boldsymbol{\theta}$ (cf. Line 4).

The following theorem gives an upper bound on the regret of Algorithm 1.

**Theorem 1** (Regret of Algorithm 1). *Suppose $K, T \leq \mathrm{poly}(d)$ and $N \geq d^2$. Under Assumption 1 and Assumption 2, the expected regret of Algorithm 1 is upper bounded by*

$$\mathbb{E}[R^{N,T}] = \widetilde{O}(T\sqrt{kN} + \sqrt{dkNT}).$$

There are two terms in Theorem 1, and we interpret them separately. The first term $\widetilde{O}(T\sqrt{kN})$ represents the regret for playing $T$ independent linear bandits *with dimension $k$* for $N$ rounds. This is the regret we need to pay even if we know the optimal feature extractor, with which we can reduce the original problem to playing $T$ independent linear bandits with dimension $k$ (recall $\boldsymbol{w}_t$ are different for different tasks). The second term $\widetilde{O}(\sqrt{dkNT})$ represents the price we need to pay to learn the feature extractor $\boldsymbol{B}$. Notably, this term shows we are using data across all tasks to learn $\boldsymbol{B}$ as this term scales with $\sqrt{NT}$.

Now comparing with the naive strategy that plays $T$ independent $d$-dimensional linear bandits with regret $\widetilde{O}(T\sqrt{dN})$, our upper bound is smaller as long as $T = \Omega(k)$. Furthermore, when $T$ is large, our bound is significantly stronger than $\widetilde{O}(T\sqrt{dN})$, especially when $k \ll d$. To our knowledge, this is the first formal theoretical result showing the advantage of representation learning for bandit problems. We remark that requiring $T = \Omega(k)$ is necessary. One needs at least $k$ tasks to recover the span of $\boldsymbol{W}$, so only in this regime representation learning can help.

Our result also puts a technical requirement on the scaling $K, T \leq \mathrm{poly}(d)$ and $N \geq d^2$. These are conditions that are often required in linear bandits literature. The first condition ensures that $K$ and $T$ are not too large, so we need not characterize $\log(KT)$ factors in regret bound. If they are too large, e.g. $K, T \geq \Omega(e^d)$, then we would have $\log KT = O(d)$ and we could no longer omit the logarithmic factors in regret bounds. The second condition ensures one can at least learn the linear coefficients up to a constant error. See more discussions in (Han et al., 2020, Section 2.5).

While Algorithm 1 is a straightforward algorithm, the proof of Theorem 1 requires a combination of representation learning and linear bandit techniques. First, we prove the in-sample guarantee of representation learning, as done in Lemma 2. Second, we exploit Assumption 2 to show that the learned parameters could extrapolate well on new contexts, as shown in Lemma 4. The regret analysis then follows naturally. We defer the proof of Theorem 1 to Appendix A.

The following theorem shows that Algorithm 1 is minimax optimal up to logarithmic factors.

**Theorem 2** (Lower Bound for Finite-Action Setting). *Let $\mathcal{A}$ denote an algorithm and $\mathcal{I}$ denote a finite-actioned multi-task linear bandit instance that satisfies Assumption 1 and Assumption 2. Then for any $N, T, d, k \in \mathbb{Z}^+$ with $k \leq d$, $k \leq T$, we have*

$$\inf_{\mathcal{A}} \sup_{\mathcal{I}} \mathbb{E}[R_{\mathcal{A},\mathcal{I}}^{N,T}] = \Omega\left(T\sqrt{kN} + \sqrt{dkNT}\right). \tag{1}$$

Theorem 2 has the exactly same two terms as in Theorem 1. This confirms our intuition that the two prices to pay are real: 1) playing $T$ independent $k$-dimensional linear bandits and 2) learning

---

**Algorithm 2:** $\mathsf{E}^2\mathsf{TC}$: Explore-Explore-Then-Commit

---

**Input:** $N$: total number of rounds , $N_1$: number of rounds for stage 1 , $N_2$: number of rounds for stage 2

1: *Stage 1: Estimating Linear Feature Extractor with Method-of-Moments*;
2: **for** $\forall t \in [T], n \in [N_1]$ **do**
3:      Play $x_{n,t} \sim \mathrm{Unif}(\lambda_0 \cdot \mathbb{S}^{d-1})$ and receive reward $r_{n,t}$;
4: Compute $\widehat{\boldsymbol{M}} \leftarrow \frac{1}{N_1 T} \sum_{n=1}^{N_1} \sum_{t=1}^{T} r_{n,t}^2 x_{n,t} x_{n,t}^\top$;
5: Let $\widehat{\boldsymbol{B}} \boldsymbol{D} \widehat{\boldsymbol{B}}^\top \leftarrow$ top-$k$ singular value decomposition of $\widehat{\boldsymbol{M}}$. Denote $\widehat{\boldsymbol{b}}_i$ the $i$-th column of $\widehat{\boldsymbol{B}}$;
6: *Stage 2: Estimating Optimal Actions on Low-dimensional Space* ;
7: **for** $\forall t \in [T], i \in [k]$ **do**
8:      Play $v_{t,i} \triangleq \sqrt{\lambda_0} \widehat{b}_i$ for $N_2/k$ times and receive rewards $\{r_{n,t}\}_{n=N_1+(i-1)N_2/k+1}^{N_1+iN_2/k}$;
9: **for** $\forall t \in [T]$ **do**
10:      Estimate $\widehat{\boldsymbol{w}}_t \leftarrow \arg \min_{\boldsymbol{w} \in \mathbb{R}^k} \frac{1}{2N_2} \sum_{n=N_1+1}^{N_1+N_2} [\langle x_{n,t}, \widehat{\boldsymbol{B}} \boldsymbol{w} \rangle - r_{n,t}]^2$;
11:      Let $\widehat{\boldsymbol{\theta}}_t = \widehat{\boldsymbol{B}} \widehat{\boldsymbol{w}}_t$;
12: *Stage 3: Committing to Near-optimal Actions*;
13: **for** $\forall t \in [T], n = N_1 + N_2 + 1, \ldots, N$ **do**
14:      Play $\boldsymbol{a}_{n,t} \leftarrow \arg \max_{\boldsymbol{a} \in \mathcal{A}_t} \langle a, \widehat{\boldsymbol{\theta}}_t \rangle$ and receive reward $r_{n,t}$;

---

the $d \times k$-dimensional feature extractor. We defer the proof of Theorem 2 to Appendix B. At a high level, we separately prove the two terms in the lower bound. The first term is established by the straightforward observation that our multi-task linear bandit problem is at least as hard as solving $T$ independent $k$-dimensional linear bandits. The second term is established by the observation that multi-task linear bandit can be seen as solving $k$ independent $d$-dimensional linear bandits, each has $NT/k$ rounds. Note that the observation would directly imply the regret lower bound $k\sqrt{d(NT/k)} = \sqrt{dkNT}$, which is exactly the second term. To our knowledge, this lower bound is also the first one of its kind for multi-task sequential decision-making problems. We believe our proof framework can be used in proving lower bounds for other related problems.

## 5    EXTENSION TO INFINITE-ACTION SETTING

In this section we present and analyze an algorithm for the infinite-action setting. Pseudo-code is listed in Algorithm 2. Our algorithm has three stages, and we explain each step below.

**Stage 1: Estimating Linear Feature Extractor with Method-of-Moments.** The goal of the first stage is to estimate the linear feature extractor $\boldsymbol{B}$. Our main idea to view this problem as a low-rank estimation problem for which we use a method-of-moments estimator. In more detail, we first sample each $x_{n,t} \sim \mathrm{Unif}[\lambda_0 \cdot \mathbb{S}^d]$ for $N_1$ times. We can use this sampling scheme because the action set is an ellipsoid. Note this sampling scheme has a sufficient coverage on all directions, which help us estimate $\boldsymbol{B}$. Next, we compute the empirical weighted covariance matrix $\widehat{\boldsymbol{M}} = \frac{1}{N_1 T} \sum_{n=1}^{N_1} \sum_{t=1}^{T} r_{n,t}^2 \boldsymbol{x}_{n,t} \boldsymbol{x}_{n,t}^\top$. To proceed, we compute the singular value decomposition of $\widehat{\boldsymbol{M}}$ and keep its top-$k$ column space as our estimated linear feature extractor $\widehat{\boldsymbol{B}}$, which is a sufficiently accurate estimator (cf. Theorem 5).

**Stage 2: Estimating Optimal Actions on Low-Dimensional Space.** In the second stage, we use our estimated linear feature extractor to refine our search space for the optimal actions. Specifically, we denote $\widehat{\boldsymbol{B}} = [\widehat{\boldsymbol{b}}_1, \ldots, \widehat{\boldsymbol{b}}_k]$ and for $i \in [k]$ and $t \in [T]$, we let $\boldsymbol{v}_{t,i} = \sqrt{\lambda_0} \widehat{\boldsymbol{b}}_i$. Under Assumption 3, we know $\boldsymbol{v}_{t,i} \in \mathcal{A}_t$ for all $i \in [k]$. Therefore, we can choose $\boldsymbol{v}_{t,i}$ to explore. Technically, our choice of actions $\boldsymbol{v}_{t,i}$ also guarantees a sufficient cover in the sense that $\lambda_{\min}\left(\sum_{i=1}^{k} \widehat{\boldsymbol{B}}^\top \boldsymbol{v}_{t,i} \boldsymbol{v}_{t,i}^\top \widehat{\boldsymbol{B}}\right) \geq \lambda_0$. In particular, this coverage is on a *low-dimensional space* instead of the original ambient space.

The second stage has $N_2$ rounds and on the $t$-th task, we just play each $\boldsymbol{v}_{t,i}$ for $N_2/k$ rounds. After that, we use linear regression to estimate $\boldsymbol{w}_t$ for each task. Given the estimation $\widehat{\boldsymbol{w}}_t$, we can obtain an estimation to the true linear coefficient $\widehat{\boldsymbol{\theta}}_t \triangleq \widehat{\boldsymbol{B}} \widehat{\boldsymbol{w}}_t$.

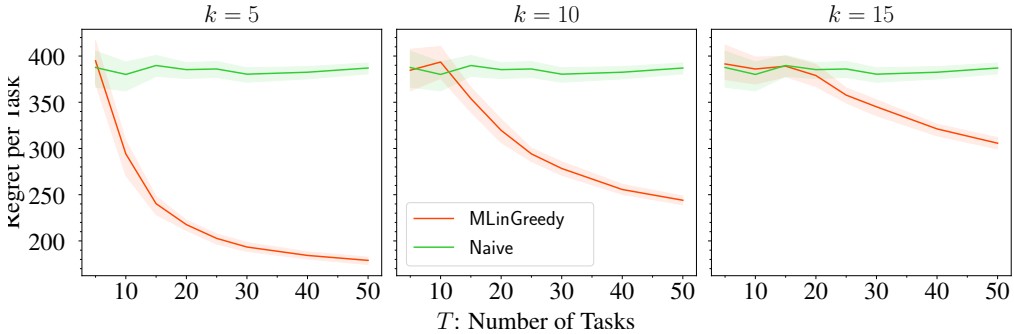

Figure 1: Comparisons of Algorithm 1 with the naive algorithm for $d = 30$ on synthetic data.

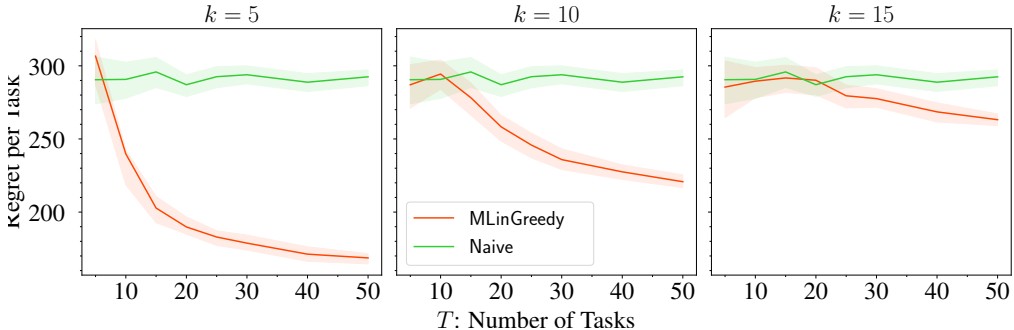

Figure 2: Comparisons of Algorithm 1 with the naive algorithm for $d = 20$ on synthetic data.

**Stage 3: Committing to Near-Optimal Actions.** After the second stage, we have an estimation $\widehat{\boldsymbol{\theta}}_t$ for each task. For the remaining $(N - N_1 - N_2)$ rounds, we just commit to the optimal action indicated by our estimations. Specifically, we play the action $\boldsymbol{a}_{n,t} \leftarrow \arg\max_{\boldsymbol{a} \in \mathcal{A}_t} \langle \boldsymbol{a}_t, \widehat{\boldsymbol{\theta}}_t \rangle$ for round $n = N_1 + N_2 + 1, \ldots, N$.

The following theorem characterizes the regret of our algorithm.

**Theorem 3** (Regret of Algorithm 2). *If we choose $N_1 = c_1 d^{1.5} k \sqrt{\frac{N}{T}}$ and $N_2 = c_2 k \sqrt{N}$ for some constants $c_1, c_2 > 0$. The regret of Algorithm 2 is upper bounded by*

$$\mathbb{E}[R^{N,T}] = \widetilde{O}(d^{1.5} k \sqrt{TN} + kT\sqrt{N}).$$

The first term represents the regret incurred by estimating the linear feature extractor. Notably, the term scales with $\sqrt{TN}$, which means we utilize all $TN$ data here. The first term scales with $d^{1.5} k$, which we conjecture is unavoidable at least by our algorithm. The second term represents playing $T$ independent $k$-dimensional infinite-action linear bandits.

Notice that if one uses a standard algorithm, e.g. the PEGE algorithm (Rusmevichientong & Tsitsiklis, 2010), to play $T$ tasks independently, one can achieve an $\widetilde{O}(dT\sqrt{N})$ regret. Comparing with this bound, our bound's second term is always smaller and the first is smaller when $T = \widetilde{\Omega}(dk^2)$. This demonstrates that more tasks indeed help us learn the representation and reduce the regret.

We complement our upper bound with a lower bound below. This theorem suggests our second term is tight but there is still a gap in the first term. We leave it as an open problem to design new algorithm to match the lower bound or to prove a stronger lower bound.

**Theorem 4** (Lower Bound for Infinite-Action Setting). *Let $\mathcal{A}$ denote an algorithm and $\mathcal{I}$ denote an infinite-action multi-task linear bandit instance that satisfies Assumption 1, Assumption 3, Assumption 4, Assumption 5. Then for any $N, T, d, k \in \mathbb{Z}^+$ with $k \leq d$, $k \leq T$, we have*

$$\inf_{\mathcal{A}} \sup_{\mathcal{I}} \mathbb{E}[R_{\mathcal{A},\mathcal{I}}^{N,T}] = \Omega\left(d\sqrt{kNT} + kT\sqrt{N}\right). \tag{2}$$

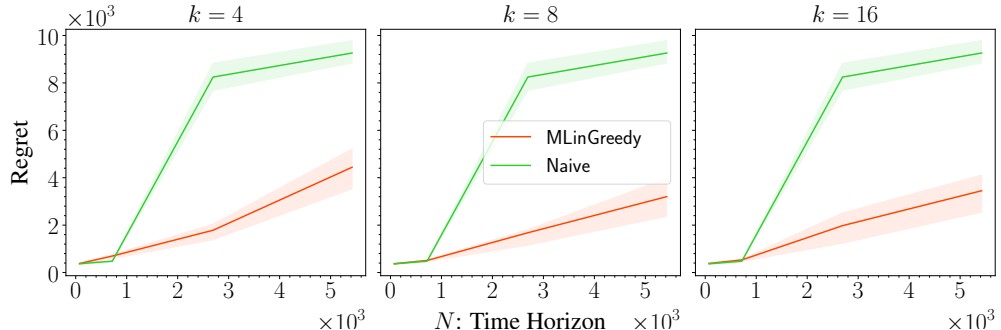

Figure 3: Comparisons of Algorithm 1 with the naive algorithm for $T = 10$ on MNIST .

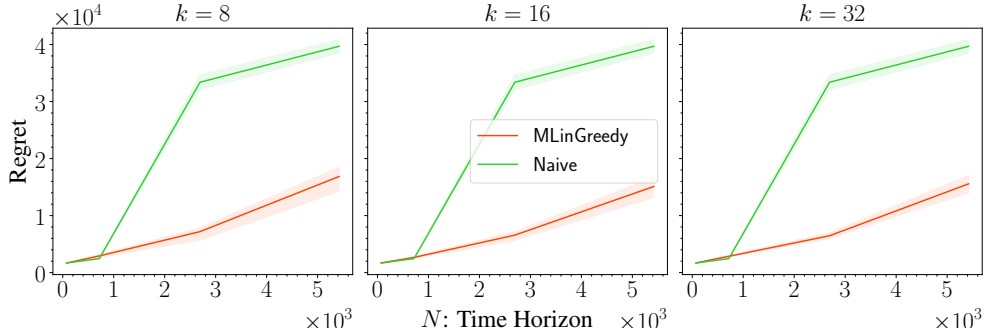

Figure 4: Comparisons of Algorithm 1 with the naive algorithm for $T = 45$ on MNIST.

## 6 EXPERIMENTS

In this section, we use synthetic data and MNIST data to illustrate our theoretical findings and demonstrate the effectiveness of our Algorithm for the finite-action setting. We also have simulation studies for the infinite-action setting, which we defer to Appendix F. The baseline is the naive algorithm which plays $T$ tasks independently, and for each task, thie algorithm uses linear regression to estimate $\boldsymbol{\theta}_t$ and choose the action greedily according to the estimated $\boldsymbol{\theta}_t$.

### 6.1 SYNTHETIC DATA

**Setup.** The linear feature extractor $\boldsymbol{B}$ is uniformly drawn from the set of $d \times k$ matrices with orthonormal columns.[2] Each linear coefficient $\boldsymbol{w}_t$ is uniformly chosen from the $k$-dimensional sphere. The noises are i.i.d. Gaussian: $\varepsilon_{n,t,a} = \mathcal{N}(0, 1)$ for every $n \in [N], t \in [T], a \in [K]$. We fix $K = 5$ and $N = 10000$ for all simulations on finite-action setting. We vary $k$, $d$ and $T$ to compare Algorithm 1 and the naive algorithm.

**Results and Discussions.** We present the simulation results in Figure 1 and Figure 2. We emphasize that the $y$-axis in our figures corresponds to the regret per task, which is defined as $R^{N,T}/T$. We fix $K = 5, N = 10000$. These simulations verify our theoretical findings. First, as the number of tasks increases, the advantage of our algorithm increases compared to the naive algorithm. Secondly, we notice that as $k$ becomes larger (relative to $d$), the advantage of our algorithm becomes smaller. This can be explained by our theorem that as $k$ increases, our algorithm pays more regret, whereas the naive algorithm's regret does not depend on $k$.

### 6.2 FINITE-ACTION LINEAR BANDITS FOR MNIST

**Setup.** We create a linear bandits problem on MNIST data (LeCun et al., 2010) to illustrate the effectiveness of our algorithm on real-world data. We fix $K = 2$ and create $T = \binom{10}{2}$ tasks and each task is parameterized by a pair $(i, j)$, where $0 \le i < j \le 9$. We use $\mathcal{D}_i$ to denote the set of MNIST images with digit $i$. At each round $n \in [N]$, for each task $(i, j)$, we randomly choose one picture

---

[2]We uniformly (under the Haar measure) choose a random element from the orthogonal group $O(d)$ and uniformly choose $k$ of its columns to generate $\boldsymbol{B}$.

from $\mathcal{D}_i$ and one from $\mathcal{D}_j$, then we present those two pictures to the algorithm and assign the picture with larger digit with reward 1 and the other with reward 0. The algorithm is now required to select an image (action). We again compare our algorithm with the naive algorithm.

**Results and Discussions.** The experimental results are displayed in Figure 3 for $T = 10$ (done by constructing tasks with first five digits) and Figure 4 for $T = 45$. We observe for both $T = 10$ and $T = 45$, our algorithm significantly outperforms the naive algorithm for all $k$. Interestingly, unlike our simulations, we find the advantage of our algorithm does not decrease as we increase $k$. We believe the reason is the optimal predictor is not exactly linear, and we need to develop an agnostic theory to explain this phenomenon, which we leave as a future work.

## 7 CONCLUSION

We initiate the study on the benefits of representation learning in bandits. We proposed new algorithms and demonstrated that in the multi-task linear bandits, if all tasks share a common linear feature extractor, then representation learning provably reduces the regret. We demonstrated empirical results to corroborate our theory. An interesting future direction is to generalize our results to general reward function classes (Li et al., 2017; Agrawal et al., 2019). In the following, we discuss some future directions.

**Adversarial Contexts** For the finite-action setting, we assumed the context are i.i.d. sampled from a Gaussian distribution. In the bandit literature, there is a large body on developing low-regret algorithms for the adversarial contexts setting. We leave it as an open problem to develop a algorithm with an $\widetilde{O}(T\sqrt{kN} + \sqrt{dkNT})$ upper bound or show this bound is not possible in the adversarial contexts setting. One central challenge for the upper bound is that existing analyses for multi-task representation learning requires i.i.d. inputs even in the supervised learning setting. Another challenge is how to develop a confidence interval for an unseen input in the multi-task linear bandits setting. This confidence interval should utilize the common feature extractor and is tighter than the standard confidence interval for linear bandits, e.g. LinUCB.

**Robust Algorithm** The current approach is tailored to the assumption that there exists a common feature extractor. One interesting direction is to develop a robust algorithm. For example, consider the scenario where whether there exists a common feature extractor is unknown. We want to develop an algorithm with regret bound as in this paper when the common feature extractor exists and gracefully degrades to the regret of $T$ independent linear bandits when the common feature extractor does not exist.

**General Function Approximation** In this paper, we focus on linear bandits. In the bandits literature, sublinear regret guarantees have been proved for more general reward function classes beyond the linear one (Li et al., 2017; Agrawal et al., 2019). Similarly, in the supervised representation learning literature, general representation function classes have also been studied (Maurer et al., 2016; Du et al., 2020). An interesting future direction is to merge these two lines of research by developing provably efficient algorithms for multi-task bandits problems where general function class is used for representation.

### ACKNOWLEDGMENTS

The authors would like to thank the anonymous reviewers for their comments and suggestions on our paper. WH is supported by NSF, ONR, Simons Foundation, Schmidt Foundation, Amazon Research, DARPA and SRC. JDL acknowledges support of the ARO under MURI Award W911NF-11-1-0303, the Sloan Research Fellowship, and NSF CCF 2002272.

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

# A  PROOF OF THEOREM 1

**Lemma 1** (General Hoeffding's inequality, Vershynin (2018), Theorem 2.6.2). *Let $X_1, \ldots, X_n$ be independent random variables such that $\mathbb{E}[X_i] = 0$ and $X_i$ is $\sigma_i$-sub-Gaussian. Then there exists a constant $c > 0$, such that for any $\delta > 0$, we have*

$$\Pr\left[\left|\sum_{i=1}^{n} X_i\right| \geq c\sqrt{\sum_{i=1}^{n} \sigma_i^2 \log(1/\delta)}\right] \leq \delta. \tag{3}$$

**Lemma 2.** *Let $T$ be the number of tasks and $N_0$ be the number of samples. For every $(n, t) \in [N_0] \times [T]$, let $\boldsymbol{x}_{n,t} \in \mathbb{R}^d$ be fixed vectors and let $y_{n,t} = \boldsymbol{x}_{n,t}^\top \boldsymbol{\theta}_t + \varepsilon_{n,t}$, where $\boldsymbol{\theta}_t \in \mathbb{R}^d$ is a vector and $\varepsilon_{n,t}$ is an independent 1-sub-Gaussian variable. Let*

$$\widehat{\boldsymbol{B}}\widehat{\boldsymbol{W}} = \underset{\widehat{\boldsymbol{B}} \in \mathbb{R}^{d \times k}, \widehat{\boldsymbol{W}} \in \mathbb{R}^{k \times T}}{\arg\min} \sum_{n=1}^{N_0} \sum_{t=1}^{T} [\boldsymbol{x}_{n,t}^\top \widehat{\boldsymbol{B}}\widehat{\boldsymbol{w}}_t - y_{n,t}]^2, \tag{4}$$

*where $\widehat{\boldsymbol{W}} = (\widehat{\boldsymbol{w}}_1 \quad \cdots \quad \widehat{\boldsymbol{w}}_T)$. Then with probability $1 - \delta$, we have*

$$\sum_{n=1}^{N_0} \sum_{t=1}^{T} [\boldsymbol{x}_{n,t}^\top (\widehat{\boldsymbol{B}}\widehat{\boldsymbol{w}}_t - \boldsymbol{B}\boldsymbol{w}_t)]^2 \lesssim (dk + kT) \log(NTdk) + \log(1/\delta).$$

*Proof.* By (4), we have

$$\sum_{n=1}^{N_0} \sum_{t=1}^{T} [\boldsymbol{x}_{n,t}^\top \widehat{\boldsymbol{B}}\widehat{\boldsymbol{w}}_t - y_{n,t}]^2 \leq \sum_{n=1}^{N_0} \sum_{t=1}^{T} [\boldsymbol{x}_{n,t}^\top \boldsymbol{B}\boldsymbol{w}_t - y_{n,t}]^2.$$

Since $y_{n,t} = \boldsymbol{x}_{n,t}^\top \boldsymbol{B}\boldsymbol{w}_t + \varepsilon_{n,t}$, we have

$$\sum_{n=1}^{N_0} \sum_{t=1}^{T} [\boldsymbol{x}_{n,t}^\top (\widehat{\boldsymbol{B}}\widehat{\boldsymbol{w}}_t - \boldsymbol{B}\boldsymbol{w}_t) + \varepsilon_{n,t}]^2 \leq \sum_{n=1}^{N_0} \sum_{t=1}^{T} \varepsilon_{n,t}^2,$$

which implies

$$\sum_{n=1}^{N_0} \sum_{t=1}^{T} [\boldsymbol{x}_{n,t}^\top (\widehat{\boldsymbol{B}}\widehat{\boldsymbol{w}}_t - \boldsymbol{B}\boldsymbol{w}_t)]^2 \leq \sum_{n=1}^{N_0} \sum_{t=1}^{T} 2\varepsilon_{n,t}\boldsymbol{x}_{n,t}^\top (\widehat{\boldsymbol{B}}\widehat{\boldsymbol{w}}_t - \boldsymbol{B}\boldsymbol{w}_t). \tag{5}$$

Next we bound the right-hand side of (5) via a uniform concentration argument. We let

$$\mathcal{B} = \{\boldsymbol{B}' \in \mathbb{R}^{d \times k} : \|\boldsymbol{B}'\|_{\max} \leq 1\}, \qquad \mathcal{W} = \{\boldsymbol{W}' \in \mathbb{R}^{k \times T} : \|\boldsymbol{W}'\|_{\max} \leq 1\}.$$

For any fixed matrices $\boldsymbol{B}' \in \mathcal{B}, \boldsymbol{W}' \in \mathcal{W}$, we write $\boldsymbol{W}' = (\boldsymbol{w}_1' \quad \cdots \quad \boldsymbol{w}_T')$ and define

$$\eta_{n,t}(\boldsymbol{B}', \boldsymbol{W}') = 2\varepsilon_{n,t}\boldsymbol{x}_{n,t}^\top (\boldsymbol{B}'\boldsymbol{w}_t' - \boldsymbol{B}\boldsymbol{w}_t).$$

Note that $\eta_{n,t}(\boldsymbol{B}', \boldsymbol{W}')$ is an independent sub-Gaussian variable with sub-Gaussian norm $2\boldsymbol{x}_{n,t}^\top (\boldsymbol{B}'\boldsymbol{w}_t' - \boldsymbol{B}\boldsymbol{w}_t)$. By the general Hoeffding's inequality (Lemma 1), with probability $1 - \delta$, we have

$$\sum_{n=1}^{N_0} \sum_{t=1}^{T} \eta_{n,t}(\boldsymbol{B}', \boldsymbol{W}') \lesssim \sqrt{f(\boldsymbol{B}', \boldsymbol{W}') \log(1/\delta)}, \tag{6}$$

$$\text{where } f(\boldsymbol{B}', \boldsymbol{W}') = \sum_{n=1}^{N} \sum_{t=1}^{T} [\boldsymbol{x}_{n,t}^\top (\boldsymbol{B}'\boldsymbol{w}_t' - \boldsymbol{B}\boldsymbol{w}_t)]^2.$$

Next we apply the $\epsilon$-net. Let $\mathcal{B}' = \mathcal{N}(\mathcal{B}, \|\cdot\|_{\max}, \epsilon), \mathcal{W}' = \mathcal{N}(\mathcal{W}, \|\cdot\|_{\max}, \epsilon)$. Applying an union bound over $\mathcal{B}' \times \mathcal{W}'$ for (6), we have

$$\Pr\left[\forall (\boldsymbol{B}', \boldsymbol{W}') \in \mathcal{B}' \times \mathcal{W}' : \sum_{n=1}^{N_0} \sum_{t=1}^{T} \eta_{n,t} \lesssim \sqrt{f(\boldsymbol{B}', \boldsymbol{W}') \log(1/\delta)}\right] \geq 1 - \delta|\mathcal{B}' \times \mathcal{W}'|.$$

Since $f(\boldsymbol{B}', \boldsymbol{W}')$ is $(3NTdk)$-Lipschitz with respect to $\|\cdot\|_{\max}$, we have

$$\Pr\left[\forall(\boldsymbol{B}', \boldsymbol{W}') \in \mathcal{B} \times \mathcal{W} : \sum_{n=1}^{N_0} \sum_{t=1}^{T} \eta_{n,t} \lesssim \sqrt{f(\boldsymbol{B}', \boldsymbol{W}') \log(1/\delta)} + 24NTdk\epsilon\right] \geq 1 - \delta|\mathcal{B}' \times \mathcal{W}'|.$$

Note that $|\mathcal{B}'| = O((1/\epsilon)^{dk})$ and $|\mathcal{W}'| = O((1/\epsilon)^{kT})$. Let $\delta = \delta_0/|\mathcal{B}' \times \mathcal{W}'|$ and $\epsilon = (24NTdk)^{-1}$. We have

$$\Pr\left[\forall(\boldsymbol{B}', \boldsymbol{W}') \in \mathcal{B} \times \mathcal{W} : \sum_{n=1}^{N_0} \sum_{t=1}^{T} \eta_{n,t} \lesssim \sqrt{f(\boldsymbol{B}', \boldsymbol{W}')[(dk + kT)\log(NTdk) + \log(1/\delta_0)]}\right] \geq 1 - \delta_0.$$
(7)

Now we assume that the above event holds. Combining (5) and (7), we have

$$\Pr\left[\sqrt{f(\widehat{\boldsymbol{B}}, \widehat{\boldsymbol{W}})} \lesssim \sqrt{(dk + kT)\log(NTdk) + \log(1/\delta_0)}\right]$$

$$= \Pr\left[f(\widehat{\boldsymbol{B}}, \widehat{\boldsymbol{W}}) \lesssim \sqrt{f(\widehat{\boldsymbol{B}}, \widehat{\boldsymbol{W}})[(dk + kT)\log(NTdk) + \log(1/\delta_0)]}\right]$$

$$\geq 1 - \delta_0,$$

which proves our lemma. $\qquad\square$

**Lemma 3.** *With probability $1 - O((NT)^{-2})$, for every $m \in [M], t \in [T]$, we have*

$$\lambda_{\min}\left(\sum_{n=\mathcal{G}_{m-1}+1}^{\mathcal{G}_m} \boldsymbol{x}_{n,t,a_{n,t}} \boldsymbol{x}_{n,t,a_{n,t}}^\top\right) \gtrsim \frac{\mathcal{G}_m - \mathcal{G}_{m-1}}{d}.$$

*Proof.* The proof can be done by following the proof of Lemma 4 in Han et al. (2020). $\qquad\square$

**Lemma 4.** *For each epoch $m \in [M]$, with probability $1 - O((NT^{-2}))$, we have*

$$\left\|\widehat{\boldsymbol{B}}\widehat{\boldsymbol{W}} - \boldsymbol{B}\boldsymbol{W}\right\|_F^2 \lesssim \frac{(dk + kT)\log(NTdk) + \log(1/\delta)}{(\mathcal{G}_m - \mathcal{G}_{m-1})/d},$$

*where $\widehat{\boldsymbol{B}}, \widehat{\boldsymbol{W}}$ are computed at Line 5 in Algorithm 1.*

*Proof.* Placing $N_0 = \mathcal{G}_m - \mathcal{G}_{m-1}$ in Lemma 2, with probability $1 - O((NT)^{-2})$, we have

$$\sum_{n=\mathcal{G}_{m-1}}^{\mathcal{G}_m} \sum_{t=1}^{T} [\boldsymbol{x}_{n,t,a_{n,t}}^\top (\widehat{\boldsymbol{B}}\widehat{\boldsymbol{w}}_t - \boldsymbol{B}\boldsymbol{w}_t)]^2 \lesssim (dk + kT)\log(NTdk).$$
(8)

By Lemma 3, with probability $1 - O((NT)^{-2})$, we have

$$\sum_{n=\mathcal{G}_{m-1}}^{\mathcal{G}_m} \sum_{t=1}^{T} [\boldsymbol{x}_{n,t,a_{n,t}}^\top (\widehat{\boldsymbol{B}}\widehat{\boldsymbol{w}}_t - \boldsymbol{B}\boldsymbol{w}_t)]^2$$

$$= \sum_{t=1}^{T} (\widehat{\boldsymbol{B}}\widehat{\boldsymbol{w}}_t - \boldsymbol{B}\boldsymbol{w}_t)^\top \left(\sum_{n=\mathcal{G}_{m-1}}^{\mathcal{G}_m} \boldsymbol{x}_{n,t,a_{n,t}} \boldsymbol{x}_{n,t,a_{n,t}}^\top\right) (\widehat{\boldsymbol{B}}\widehat{\boldsymbol{w}}_t - \boldsymbol{B}\boldsymbol{w}_t)$$

$$\gtrsim \sum_{t=1}^{T} (\widehat{\boldsymbol{B}}\widehat{\boldsymbol{w}}_t - \boldsymbol{B}\boldsymbol{w}_t)^\top \frac{\mathcal{G}_m - \mathcal{G}_{m-1}}{d} (\widehat{\boldsymbol{B}}\widehat{\boldsymbol{w}}_t - \boldsymbol{B}\boldsymbol{w}_t)$$

$$= \frac{\mathcal{G}_m - \mathcal{G}_{m-1}}{d} \left\|\widehat{\boldsymbol{B}}\widehat{\boldsymbol{W}} - \boldsymbol{B}\boldsymbol{W}\right\|_F^2.$$
(9)

We conclude by combining (9) with (8). $\qquad\square$

Let

$$R_m = \sum_{n=\mathcal{G}_{m-1}+1}^{\mathcal{G}_m} \sum_{t=1}^{T} \max_{a \in [K]} \langle \boldsymbol{x}_{n,t,a}, \boldsymbol{\theta}_t \rangle - \langle \boldsymbol{x}_{n,t,a_{n,t}}, \boldsymbol{\theta}_t \rangle$$

be the regret incurred in the $m$-th epoch. We have the following lemma.

**Lemma 5.** *We have*

$$\mathbb{E}[R_m] \lesssim (\sqrt{NTdk} + T\sqrt{kN})\sqrt{\log(NTdk)\log(NKT)}.$$

*Proof.* At round $n$ that belongs to epoch $m$, for task $t$, we have

$$\max_{a \in [K]} \boldsymbol{\theta}_t^\top (\boldsymbol{x}_{n,t,a} - \boldsymbol{x}_{n,t,a_{n,t}}) \leq \max_{a \in [K]} \{\boldsymbol{\theta}_t^\top (\boldsymbol{x}_{n,t,a} - \boldsymbol{x}_{n,t,a_{n,t}}) + \widehat{\boldsymbol{\theta}}_{m-1,t}^\top (\boldsymbol{x}_{n,t,a_{n,t}} - \boldsymbol{x}_{n,t,a})\}$$

$$= \max_{a \in [K]} (\boldsymbol{\theta}_t - \widehat{\boldsymbol{\theta}}_{m-1,t})^\top \boldsymbol{x}_{n,t,a} + (\widehat{\boldsymbol{\theta}}_{m-1,t} - \boldsymbol{\theta}_t)^\top \boldsymbol{x}_{n,t,a_{n,t}}$$

$$\leq 2 \max_{a \in [K]} \left| (\boldsymbol{\theta}_t - \widehat{\boldsymbol{\theta}}_{m-1,t})^\top \boldsymbol{x}_{n,t,a} \right|.$$

By Assumption 2 and a union bound over $K$ actions, $T$ tasks, and $(\mathcal{G}_m - \mathcal{G}_{m-1})$ rounds, we have with probability $1 - (NKT)^{-2}$, for every $n \in (\mathcal{G}_{m-1}, \mathcal{G}_m]$ and $t \in [T]$,

$$\max_{a \in [K]} \left| (\boldsymbol{\theta}_t - \widehat{\boldsymbol{\theta}}_{m-1,t})^\top \boldsymbol{x}_{n,t,a} \right| \lesssim \left\| \boldsymbol{\theta}_t - \widehat{\boldsymbol{\theta}}_{m-1,t} \right\| \sqrt{\frac{\log(NKT)}{d}}. \tag{10}$$

Using a union bound over (10) and Lemma 4, with probability $1 - O((NT)^{-2})$, the regret incurred in the $m$-th epoch is

$$\mathbb{E}[R_m] \lesssim \sum_{n=\mathcal{G}_{m-1}+1}^{\mathcal{G}_m} \sum_{t=1}^{T} \max_{a \in [K]} \boldsymbol{\theta}_t^\top (\boldsymbol{x}_{n,t,a} - \boldsymbol{x}_{n,t,a_{n,t}})$$

$$\lesssim (\mathcal{G}_m - \mathcal{G}_{m-1}) \sum_{t=1}^{T} \left\| \boldsymbol{\theta}_t - \widehat{\boldsymbol{\theta}}_{m-1,t} \right\| \sqrt{\frac{\log(NKT)}{d}} \leq \mathcal{G}_m \sqrt{T} \left\| \widehat{\boldsymbol{\Theta}} - \boldsymbol{\Theta} \right\|_F \sqrt{\frac{\log(NKT)}{d}} \tag{11}$$

$$\leq \mathcal{G}_m \sqrt{T} \sqrt{\frac{(dk + kT)\log(NTdk)}{(\mathcal{G}_{m-1} - \mathcal{G}_{m-2})/d}} \sqrt{\frac{\log(NKT)}{d}} \tag{12}$$

$$\lesssim \sqrt{NT(dk + kT)\log(NTdk)\log(NKT)} \tag{13}$$

$$\lesssim (\sqrt{NTdk} + T\sqrt{kN})\sqrt{\log(NTdk)\log(NKT)}, \qquad \square$$

where we denote $\widehat{\boldsymbol{\Theta}} = \begin{pmatrix} \widehat{\boldsymbol{\theta}}_{m-1,1} & \cdots & \widehat{\boldsymbol{\theta}}_{m-1,T} \end{pmatrix}$ in (11) and the second inequality in (11) uses Cauchy, (12) uses Lemma 4, (13) uses $\mathcal{G}_{m-1} \lesssim \mathcal{G}_{m-1} - \mathcal{G}_{m-2}$ and $\mathcal{G}_m/\mathcal{G}_{m-1} \lesssim \sqrt{N}$. Since that the above bound holds with probability $1 - O((NT)^{-2})$ and that the regret is bounded by $NT$, we prove the lemma.

*Proof of Theorem 1.* The regret is bounded by

$$\mathbb{E}[R^{N,T}] = \sum_{m=1}^{M} \mathbb{E}[R_m] \lesssim M(\sqrt{NTdk} + T\sqrt{kN})\sqrt{\log(NTdk)\log(NKT)}$$

$$= (\sqrt{NTdk} + T\sqrt{kN})\sqrt{\log(NTdk)\log(NKT)} \log \log N. \qquad \square$$

## B    PROOF OF THEOREM 2

In this appendix, we assume that $\Sigma_t = \boldsymbol{I}$ in Assumption 2 and that all noises are gaussian, i.e. $\varepsilon_{n,t} \sim \mathcal{N}(0, 1)$ for all $n \in [N], t \in [T]$.

For each task $t \in [T]$, we denote the regret incurred on task $t$ as

$$R^{N,(t)} = \sum_{n=1}^{N} \max_{a \in [K]} \langle \boldsymbol{x}_{n,t,a}, \boldsymbol{\theta}_t \rangle - \langle \boldsymbol{x}_{n,t,a_{n,t}}, \boldsymbol{\theta}_t \rangle.$$

We divide Theorem 2 into the following two lemmas.

**Lemma 6.** *Under the setting of Theorem 2, we have* $\inf_{\mathcal{A}} \sup_{\mathcal{I}} \mathbb{E}[R_{\mathcal{A},\mathcal{I}}^{N,T}] \geq \Omega(T\sqrt{kN})$.

**Lemma 7.** *Under the setting of Theorem 2, we have* $\inf_{\mathcal{A}} \sup_{\mathcal{I}} \mathbb{E}[R_{\mathcal{A},\mathcal{I}}^{N,T}] \geq \Omega(\sqrt{dkNT})$.

*Proof of Theorem 2.* We combine Lemma 6 and Lemma 7. $\qquad\square$

Our proofs to the lemmas will be based on the lower bounds for the (single-task) linear bandit setting, which corresponds to the $T = 1$ case in our multi-task setting. For this single-task setting, we assume $k = d$ and $\boldsymbol{B} = \boldsymbol{I}_d$. We write the regret as $R^N = R^{N,1}$ and call algorithms for the single-task setting as single-task algorithms.

**Lemma 8** (Han et al. (2020), Theorem 2). *Assume $N \geq d^2$ and $d \geq 2$. Let $\mathcal{N}(\mu, \Sigma)$ be the multivariate normal distribution with mean $\mu$ and covariance matrix $\Sigma$. There is a constant $C > 0$, such that for any single-task algorithm $\mathcal{S}$, we have*

$$\sup_{\|\boldsymbol{w}\| \leq 1} \mathbb{E}[R_{\mathcal{S},\mathcal{I}}^N] \geq C\sqrt{dN},$$

*where $\mathcal{I}$ is the instance with hidden linear coefficients $\boldsymbol{w}$.*

Next we use it to prove Lemma 6 and Lemma 7. The main idea to prove Lemma 6 is to note that we can treat our setting as $T$ independent $k$-dimensional linear bandits.

*Proof of Lemma 6.* Suppose there is an algorithm $\mathcal{A}$ that achieves $\sup_{\mathcal{I}} \mathbb{E}[R_{\mathcal{A},\mathcal{I}}^{N,T}] \leq CT\sqrt{kN}$. Then we have

$$\sup_{\|\boldsymbol{w}_t\| \leq 1} \mathbb{E}\left[\sum_{t=1}^{T} R_{\mathcal{A},\mathcal{I}}^{N,(t)}\right] \leq \sup_{\mathcal{I}} \mathbb{E}[R_{\mathcal{A},\mathcal{I}}^{N,T}] \leq CT\sqrt{kN}.$$

Therefore, there exists $t \in [T]$ such that

$$\sup_{\mathcal{I}} \mathbb{E}[R_{\mathcal{A},\mathcal{I}}^{N,(t)}] \leq \frac{1}{T}CT\sqrt{kN} = C\sqrt{kN},$$

which contradicts Lemma 8. $\qquad\square$

*Proof of Lemma 7.* Suppose there is an algorithm $\mathcal{A}$ that achieves $\sup_{\mathcal{I}} \mathbb{E}[R_{\mathcal{A},\mathcal{I}}^{N,T}] \leq C\sqrt{dkNT}$. We complete the proof separately, based on whether $k \leq \frac{d}{2}$ or not. Note that when $k > \frac{d}{2}$, the lower bound in Lemma 6 becomes $\Omega(T\sqrt{kN}) = \Omega(T\sqrt{dN})$. Since $T \geq k$, we have $\sqrt{dkTN} \lesssim T\sqrt{dN}$. Thus we conclude by Lemma 6.

In the remaining, we assume $k \leq \frac{d}{2}$. Without loss of generality, we assume that $d$ is even and that $2k$ divides $T$. For $i = 1, \ldots, k$, we denote the regret of group $i$ as

$$R_{\mathcal{A},\mathcal{I}}^{N,((i))} = \sum_{t=(i-1)T/k+1}^{iT/k} R_{\mathcal{A},\mathcal{I}}^{N,(t)}.$$

We consider instances such that tasks from the same group share the same hidden linear coefficients $\boldsymbol{\theta}_t$. Since there are $k$ groups, we have

$$\sup_{\|\boldsymbol{\theta}_t\| \leq 1} \mathbb{E}\left[\sum_{i=1}^{k} R_{\mathcal{A},\mathcal{I}}^{N,((i))}\right] \leq \sup_{\mathcal{I}} \mathbb{E}[R_{\mathcal{A},\mathcal{I}}^{N,T}] \leq C\sqrt{dkNT}. \tag{14}$$

Therefore, there exists a group $i \in [k]$ such that

$$\sup_{\mathcal{I}_i} \mathbb{E}[R_{\mathcal{A},\mathcal{I}}^{N,((i))}] \leq \frac{1}{k} C \sqrt{dkNT} = C \sqrt{\frac{dNT}{k}}, \tag{15}$$

which means that the regret incurred in group $i$ is less than $C\sqrt{dNT/k}$. Since the tasks in group $i$ share the same hidden linear coefficients, they could be regarded as one large linear bandit problem. Since there are $T/k$ tasks in group $i$, the large linear bandit is played for $N \cdot T/k$ rounds. By Lemma 8, the algorithm $\mathcal{A}$ must have incurred $C\sqrt{dNT/k}$ regret on group $i$, which contradicts with (15). $\qquad\square$

## C  METHOD-OF-MOMENTS ESTIMATOR UNDER BANDIT SETTING

The following theorem shows the guarantee of the method-of-moments (MoM) estimator we used to find the linear feature extractor $\boldsymbol{B}$. In this appendix, for a matrix $\boldsymbol{B}$ with orthogonal columns, we write $\boldsymbol{B}_\perp$ to denote its orthogonal complement matrix (a matrix whose columns are the orthogonal complement of those of $\boldsymbol{B}$).

**Theorem 5** (MoM Estimator). *Assume $N_1 T \gtrsim \mathrm{polylog}(N_1, T) \cdot \frac{d^{1.5}k}{\lambda_0 \nu}$. We have with probability at least $1 - (N_1 T)^{-100}$,*

$$\left\| \widehat{\boldsymbol{B}}_\perp^\top \boldsymbol{B} \right\| \lesssim \widetilde{O}\left(\frac{d^{1.5}k}{\lambda_0 \nu \sqrt{N_1 T}}\right). \tag{16}$$

The theorem guarantees that our estimated $\widehat{\boldsymbol{B}}$ is close to the underlying $\boldsymbol{B}$ in the operator norm so long as the values $N_1$ and $T$ are sufficiently large. We add a remark that our theorem is similar to Theorem 3 in Tripuraneni et al. (2020). The key differences are: (i) we use a uniform distribution to find the feature extractor, while they assumed the input distribution is standard $d$-dimensional Gaussian; (ii) the SNR (signal-to-noise ratio) in our linear bandit setting is worse than that in their supervised learning setting, and thus we get an extra $d$ factor in our theorem.

In the sequel, we prove the theorem.

**Lemma 9** (Hoeffding). *Let $\varepsilon_1, \ldots, \varepsilon_N$ be i.i.d. $1$-sub-Gaussian random variables. We have*

$$\Pr\left[\left|\frac{1}{N}\sum_{i=1}^N (\varepsilon_i - \mathbb{E}\,\varepsilon_i)\right| \geq t\right] \leq 2e^{-\frac{t^2}{2N}}.$$

**Lemma 10** (Matrix Bernstein's inequality, Vershynin (2018), Theorem 5.4.1). *Let $\boldsymbol{X}_1, \ldots, \boldsymbol{X}_m \in \mathbb{R}^{d \times d}$ be independent, mean zero, symmetric random matrices that $\|\boldsymbol{X}_i\| \leq M$ almost surely for all $i \in [m]$. Let $\sigma^2 = \left\|\sum_{i=1}^m \mathbb{E}\,\boldsymbol{X}_i^2\right\|$. We have*

$$\Pr\left[\left\|\sum_{i=1}^m \boldsymbol{X}_i\right\| \geq \delta\right] \leq 2d\exp\left(-\frac{\delta^2/2}{\sigma^2 + M\delta/3}\right).$$

*Equivalently, with probability at least $1 - \delta$, we have*

$$\left\|\sum_{i=1}^m \boldsymbol{X}_i\right\| \lesssim \sqrt{\sigma^2 \log(d/\delta)} + M\log(d/\delta).$$

**Lemma 11** (Moments of Uniform Distribution on Sphere). *Let $\boldsymbol{x} \sim \mathrm{Unif}(\mathbb{S}^{d-1})$ be a uniformly chosen unit vector. We have $\mathbb{E}\,x_1^6 = \frac{15}{d(d+2)(d+4)}$, $\mathbb{E}\,x_1^4 = \frac{3}{d(d+2)}$, $\mathbb{E}\,x_1^2 = \frac{1}{d}$. Moreover, we have $\mathbb{E}\,x_1^4 x_2^2 = \frac{3}{d(d+2)(d+4)}$.*

*Proof.* We need to recall the fact that when $\boldsymbol{x} \sim \mathrm{Unif}(\mathbb{S}^{d-1})$, its coordinate $x_i$ follows the Beta distribution: $\frac{x_1+1}{2} \sim \mathrm{Beta}(\frac{d-1}{2}, \frac{d-1}{2})$. Then we prove the lemma by noting the moments of the Beta distribution (Fang, 2018). $\qquad\square$

**Corollary 6** (Uniform Distribution on Sphere). *Let $x \sim \mathrm{Unif}(\mathbb{S}^{d-1})$ be a uniformly chosen unit vector. We have the following statements.*

*(a)* $\mathbb{E} \langle \boldsymbol{x}, \boldsymbol{\theta} \rangle^2 \boldsymbol{x}\boldsymbol{x}^\top = \frac{2\boldsymbol{\theta}\boldsymbol{\theta}^\top + \boldsymbol{I}}{d(d+2)}$.

*(b)* $\mathbb{E} \langle \boldsymbol{x}, \boldsymbol{\theta} \rangle^4 \boldsymbol{x}\boldsymbol{x}^\top = \frac{12\boldsymbol{\theta}\boldsymbol{\theta}^\top + 3\boldsymbol{I}}{d(d+2)(d+4)}$.

*Proof.* Let $\boldsymbol{e}_1 = (1, 0, \ldots, 0) \in \mathbb{R}^d$ be the unit vector. For (a), note that

$$
(\mathbb{E} \langle \boldsymbol{x}, \boldsymbol{e}_1 \rangle^2 \boldsymbol{x}\boldsymbol{x}^\top)_{ij} = \mathbb{E} \, x_1^2 x_i x_j = \begin{cases} 0, & i, j \neq 1 \text{ and } i \neq j, \\ 0, & i = 1 \neq j \text{ or } j = 1 \neq i, \\ \frac{1}{d(d+2)}, & i = j \neq 1, \\ \frac{3}{d(d+2)}, & i = j = 1. \end{cases}
$$

Therefore, we have $\mathbb{E}[\langle \boldsymbol{x}, \boldsymbol{e}_1 \rangle^2 \boldsymbol{x}\boldsymbol{x}^\top] = \frac{1}{d(d+2)}(2\boldsymbol{e}_1\boldsymbol{e}_1^\top + \boldsymbol{I})$. By the isotropy (rotation invariance) of uniform distribution, we have

$$
\mathbb{E} \langle \boldsymbol{x}, \boldsymbol{\theta} \rangle^2 \boldsymbol{x}\boldsymbol{x}^\top = \frac{1}{d(d+2)}(2\boldsymbol{\theta}\boldsymbol{\theta}^\top + \boldsymbol{I}).
$$

For (b), note that

$$
(\mathbb{E} \langle \boldsymbol{x}, \boldsymbol{e}_1 \rangle^4 \boldsymbol{x}\boldsymbol{x}^\top)_{ij} = \mathbb{E} \, x_1^4 x_i x_j = \begin{cases} 0, & i, j \neq 1 \text{ and } i \neq j, \\ 0, & i = 1 \neq j \text{ or } j = 1 \neq i, \\ \frac{3}{d(d+2)(d+4)}, & i = j \neq 1, \\ \frac{15}{d(d+2)(d+4)}, & i = j = 1. \end{cases}
$$

Therefore, we have $\mathbb{E} \langle \boldsymbol{x}, \boldsymbol{e}_1 \rangle^4 \boldsymbol{x}\boldsymbol{x}^\top = \frac{1}{d(d+2)(d+4)}(12\boldsymbol{e}_1\boldsymbol{e}_1^\top + 3\boldsymbol{I})$. By the isotropy, we have

$$
\mathbb{E} \langle \boldsymbol{x}, \boldsymbol{\theta} \rangle^4 \boldsymbol{x}\boldsymbol{x}^\top = \frac{1}{d(d+2)(d+4)}(12\boldsymbol{\theta}\boldsymbol{\theta}^\top + 3\boldsymbol{I}). \qquad \square
$$

Let $\boldsymbol{A}_{n,t} = r_{n,t}^2 x_{n,t} x_{n,t}^\top, \boldsymbol{M} = \frac{1}{N_1 T} \sum_{n=1}^{N_1} \sum_{t=1}^T \boldsymbol{A}_{n,t}$. We decompose $\boldsymbol{M}$ into three terms $\boldsymbol{M} = \boldsymbol{M}_1 + \boldsymbol{M}_2 + \boldsymbol{M}_3$, where

$$
\boldsymbol{M}_1 = \frac{1}{N_1 T} \sum_{n=1}^{N_1} \sum_{t=1}^T \langle \boldsymbol{x}_{n,t}, \boldsymbol{\theta}_t \rangle^2 \boldsymbol{x}_{n,t} \boldsymbol{x}_{n,t}^\top,
$$

$$
\boldsymbol{M}_2 = \frac{1}{N_1 T} \sum_{n=1}^{N_1} \sum_{t=1}^T 2\varepsilon_{n,t} \langle \boldsymbol{x}_{n,t}, \boldsymbol{\theta}_t \rangle \boldsymbol{x}_{n,t} \boldsymbol{x}_{n,t}^\top,
$$

$$
\boldsymbol{M}_3 = \frac{1}{N_1 T} \sum_{n=1}^{N_1} \sum_{t=1}^T \varepsilon_{n,t}^2 \boldsymbol{x}_{n,t} \boldsymbol{x}_{n,t}^\top.
$$

Let $\boldsymbol{A}_{1nt} = \frac{1}{N_1 T} \langle \boldsymbol{x}_{n,t}, \boldsymbol{\theta}_t \rangle^2 \boldsymbol{x}_{n,t} \boldsymbol{x}_{n,t}^\top$. Next we analyze each error $\|\boldsymbol{M}_i - \mathbb{E}\, \boldsymbol{M}_i\|$.

**Lemma 12.** *With probability at least $1 - \frac{1}{N_1^3 T^3}$, we have*

$$
\|\boldsymbol{M}_1 - \mathbb{E}\, \boldsymbol{M}_1\| \lesssim \sqrt{\frac{\log(dN_1 T)}{d^3 N_1 T}} + \frac{\log(dN_1 T)}{N_1 T}.
$$

*Proof.* We have

$$
\mathbb{E}\, \boldsymbol{A}_{1nt}^2 = \frac{1}{N_1^2 T^2} \mathbb{E} \langle \boldsymbol{x}_{n,t}, \boldsymbol{\theta}_t \rangle^4 \boldsymbol{x}_{n,t} \boldsymbol{x}_{n,t}^\top = \frac{12\boldsymbol{\theta}_t \boldsymbol{\theta}_t^\top + 3\boldsymbol{I}}{d(d+2)(d+4)N_1^2 T^2}.
$$

Using Lemma 10 with $m = N_1 T, M = \frac{1}{N_1 T}, \sigma^2 \lesssim m \cdot \frac{1}{d^3 N_1^2 T^2} = \frac{1}{d^3 N_1 T}$, we have with probability at least $1 - \frac{1}{N_1^3 T^3}$,

$$
\|\boldsymbol{M}_1 - \mathbb{E}\, \boldsymbol{M}_1\| \lesssim \sqrt{\frac{\log(dN_1 T)}{d^3 N_1 T}} + \frac{\log(dN_1 T)}{N_1 T}. \qquad \square
$$

Next we analyze the errors of $\boldsymbol{M}_2$ and $\boldsymbol{M}_3$. Since these errors contain the unbounded sub-Gaussian terms $\varepsilon_{n,t}$, we need to cut their tails before applying the matrix Bernstein inequality. Define $\varepsilon'_{n,t} = \varepsilon_{n,t}\,\mathbb{I}\{|\varepsilon_{n,t}| \leq R\}$. We have

$$
\begin{aligned}
\left|\mathbb{E}\,\varepsilon_{n,t} - \mathbb{E}\,\varepsilon'_{n,t}\right| &\leq \mathbb{E}\,|\varepsilon_{n,t}|\,\mathbb{I}\{|\varepsilon_{n,t}| > R\} \\
&= R \cdot \Pr[|\varepsilon_{n,t}| > R] + \int_R^{+\infty} \Pr[|\varepsilon_{n,t}| > x]\,\mathrm{d}x \\
&\leq 2Re^{-\frac{R^2}{2}} + \int_R^{+\infty} 2e^{-\frac{x^2}{2}}\,\mathrm{d}x \\
&\leq 2(R + \frac{1}{R})e^{-\frac{R^2}{2}}.
\end{aligned}
$$

Let $\varepsilon''_{n,t} = \varepsilon_{n,t}^2\,\mathbb{I}\{|\varepsilon_{n,t}| \leq R\}$. We have

$$
\begin{aligned}
\left|\mathbb{E}\,\varepsilon_{n,t}^2 - \mathbb{E}\,\varepsilon''_{n,t}\right| &\leq \mathbb{E}\,\varepsilon_{n,t}^2\,\mathbb{I}\{\varepsilon_{n,t}^2 > R^2\} \\
&= R^2 \cdot \Pr[\varepsilon_{n,t}^2 > R^2] + \int_{R^2}^{+\infty} \Pr(\varepsilon_{n,t}^2 > x)\,\mathrm{d}\,\mathrm{d}x \\
&\leq 2R^2 e^{-\frac{R^2}{2}} + \int_{R^2}^{+\infty} 2e^{-\frac{x}{2}}\,\mathrm{d}\,\mathrm{d}x \\
&\leq (2R^2 + 4)e^{-\frac{R^2}{2}}.
\end{aligned}
$$

Define

$$
\boldsymbol{M}'_2 = \frac{1}{N_1 T}\sum_{n=1}^{N_1}\sum_{t=1}^T 2\varepsilon'_{n,t}\langle\boldsymbol{x}_{n,t},\boldsymbol{\theta}_t\rangle\boldsymbol{x}_{n,t}\boldsymbol{x}_{n,t}^\top,
$$

$$
\boldsymbol{M}''_3 = \frac{1}{N_1 T}\sum_{n=1}^{N_1}\sum_{t=1}^T \varepsilon''_{n,t}\boldsymbol{x}_{n,t}\boldsymbol{x}_{n,t}^\top.
$$

**Lemma 13.** *With probability at least $1 - (N_1 T)^{-3}$, we have*

$$
\|\boldsymbol{M}'_2 - \mathbb{E}\,\boldsymbol{M}'_2\| \lesssim \sqrt{\frac{\log(dN_1 T)}{d^2 N_1 T}} + \frac{R\log(dN_1 T)}{N_1 T}.
$$

*Proof.* Let $\boldsymbol{A}_{2nt} = \frac{1}{N_1 T}2\varepsilon'_{n,t}\langle\boldsymbol{x}_{n,t},\boldsymbol{\theta}_t\rangle\boldsymbol{x}_{n,t}\boldsymbol{x}_{n,t}^\top$. We find that

$$
\mathbb{E}\,\boldsymbol{A}_{2nt}^2 = \frac{4\,\mathbb{E}\,(\varepsilon'_{n,t})^2}{N_1^2 T^2}\,\mathbb{E}\,\langle\boldsymbol{x}_{n,t},\boldsymbol{\theta}_t\rangle^2\boldsymbol{x}_{n,t}\boldsymbol{x}_{n,t}^\top = \frac{4\,\mathbb{E}\,(\varepsilon'_{n,t})^2}{N_1^2 T^2}\frac{2\boldsymbol{\theta}_t\boldsymbol{\theta}_t^\top + \boldsymbol{I}}{d(d+2)}.
$$

We conclude by using Lemma 10 with $m = N_1 T$, $M = \frac{R}{N_1 T}$, $\sigma^2 \lesssim m \cdot \frac{1}{d^2 N_1^2 T^2} = \frac{1}{d^2 N_1 T}$. $\qquad\square$

**Lemma 14.** *With probability at least $1 - (N_1 T)^{-3}$, we have*

$$
\|\boldsymbol{M}''_3 - \mathbb{E}\,\boldsymbol{M}''_3\| \lesssim \sqrt{\frac{\log(dN_1 T)}{dN_1 T}} + \frac{R^2\log(dN_1 T)}{N_1 T}.
$$

*Proof.* Let $\boldsymbol{A}_{3nt} = \frac{1}{N_1 T}\varepsilon''_{n,t}\boldsymbol{x}_{n,t}\boldsymbol{x}_{n,t}^\top$. We find that

$$
\mathbb{E}\,\boldsymbol{A}_{3nt}^2 = \frac{\mathbb{E}\,(\varepsilon''_{n,t})^2}{N_1^2 T^2}\,\mathbb{E}\,\boldsymbol{x}_{n,t}x_{n,t}^\top = \frac{\mathbb{E}\,(\varepsilon''_{n,t})^2}{N_1^2 T^2}\frac{\boldsymbol{I}}{d}.
$$

We conclude by using Lemma 10 with $m = N_1 T$, $M = \frac{R^2}{N_1 T}$, $\sigma^2 \lesssim m \cdot \frac{1}{dN_1^2 T^2} = \frac{1}{dN_1 T}$. $\qquad\square$

**Lemma 15.** *With probability at least $1 - (N_1 T)^{-2}$, we have*

$$
\|\boldsymbol{M} - \mathbb{E}\,\boldsymbol{M}\| \lesssim \sqrt{\frac{\log(dN_1 T)}{dN_1 T}} + \frac{\log^2(dN_1 T)}{N_1 T}.
$$

*Proof.* Let $R = \sqrt{8 \log(N_1 T)}$. With probability at least $1 - (N_1 T)^{-3}$, we have $|\varepsilon_{n,t}| \leq R$ for every $n \in [N_1]$ and $t \in [T]$. Note that in this case, we have $\boldsymbol{M}_2 = \boldsymbol{M}_2'$ and $\boldsymbol{M}_3 = \boldsymbol{M}_3''$. Using a union bound over Lemma 12, Lemma 13, and Lemma 14, we have with probability at least $1 - (N_1 T)^{-2}$,

$$
\begin{aligned}
\|\boldsymbol{M} - \mathbb{E}\,\boldsymbol{M}\| &\leq \|\boldsymbol{M}_1 - \mathbb{E}\,\boldsymbol{M}_1\| + \|\boldsymbol{M}_2 - \mathbb{E}\,\boldsymbol{M}_2\| + \|\boldsymbol{M}_3 - \mathbb{E}\,\boldsymbol{M}_3\| \\
&= \|\boldsymbol{M}_1 - \mathbb{E}\,\boldsymbol{M}_1| + \|\boldsymbol{M}_2' - \mathbb{E}\,\boldsymbol{M}_2\| + \|\boldsymbol{M}_3'' - \mathbb{E}\,\boldsymbol{M}_3\| \\
&\leq \|\boldsymbol{M}_1 - \mathbb{E}\,\boldsymbol{M}_1\| + \|\boldsymbol{M}_2' - \mathbb{E}\,\boldsymbol{M}_2'\| + \|\mathbb{E}\,\boldsymbol{M}_2' - \mathbb{E}\,\boldsymbol{M}_2\| \\
&\quad + \|\boldsymbol{M}_3'' - \mathbb{E}\,\boldsymbol{M}_3''\| + \|\mathbb{E}\,\boldsymbol{M}_3'' - \mathbb{E}\,\boldsymbol{M}_3\| \\
&\lesssim \sqrt{\frac{\log(dN_1 T)}{dN_1 T}} + \frac{\log^2(dN_1 T)}{N_1 T}. \qquad\qquad \square
\end{aligned}
$$

*Proof of Theorem 5.* We note that $\sigma_{k+1}(\boldsymbol{M}) - \sigma_{k+1}(\mathbb{E}\,\boldsymbol{M}) \leq \|\mathbf{E}\|$. Under Assumption 3, we have

$$
\mathbb{E}\,\boldsymbol{M}_1 = \frac{2\lambda_0}{d(d+2)T}\boldsymbol{\theta}\boldsymbol{\theta}^\top + c_1\mathbf{I}, \quad \mathbb{E}\,\boldsymbol{M}_2 = 0, \quad \mathbb{E}\,\boldsymbol{M}_3 = c_3\mathbf{I},
$$

where $c_1, c_3 \in \mathbb{R}$ are constants. Since

$$
\sigma_k(\frac{1}{T}\boldsymbol{\theta}\boldsymbol{\theta}^\top) - \sigma_{k+1}(\frac{1}{T}\boldsymbol{\theta}\boldsymbol{\theta}^\top) = \sigma_k(\frac{1}{T}\mathbf{W}\mathbf{W}^\top) = \frac{\nu}{k},
$$

we have

$$
\sigma_k(\mathbb{E}\,\boldsymbol{M}) - \sigma_{k+1}(\mathbb{E}\,\boldsymbol{M}) \asymp \frac{\lambda_0\nu}{d^2 k}.
$$

Assume $N_1 T \gtrsim \text{polylog}(N_1, T) \cdot dk$ so that $\|\mathbf{E}\| \leq \frac{\lambda_0\nu}{d^2 k}$. Together with Davis-Kahan sin $\boldsymbol{\theta}$ theorem, we have with probability at least $1 - (N_1 T)^{-100}$,

$$
\begin{aligned}
\|\widehat{\boldsymbol{B}}_\perp^\top \boldsymbol{B}\| &\lesssim \frac{\|\widehat{\boldsymbol{B}}_\perp^\top \mathbf{E}\boldsymbol{B}\|}{\sigma_k(\mathbb{E}\,\boldsymbol{M}) - \sigma_{k+1}(\mathbb{E}\,\boldsymbol{M}) - \|\mathbf{E}\|} \qquad\qquad\qquad (17) \\
&\leq \frac{\|\mathbf{E}\|}{\sigma_k(\mathbb{E}\,\boldsymbol{M}) - \sigma_{k+1}(\mathbb{E}\,\boldsymbol{M}) - \|\mathbf{E}\|} \lesssim \frac{\|\mathbf{E}\|}{\lambda_0\nu/d^2 k} \\
&\lesssim \frac{d^2 k}{\lambda_0\nu}\left(\sqrt{\frac{\log(dN_1 T)}{dN_1 T}} + \frac{\log^2(dN_1 T)}{N_1 T}\right) \lesssim \frac{d^{1.5}k}{\sqrt{N_1 T}} \cdot \text{polylog}(d, N, T),
\end{aligned}
$$

where (17) uses the Davis-Kahan sin $\boldsymbol{\theta}$ theorem (Bhatia, 2013, Section VII.3). $\qquad \square$

## D  PROOF OF THEOREM 3

Our proof is similar to the proof of Theorem 3.1 of Rusmevichientong & Tsitsiklis (2010).

**Lemma 16** (Rusmevichientong & Tsitsiklis (2010), Lemma 3.5). *For two vectors $\boldsymbol{u}, \boldsymbol{v} \in \mathbb{R}^d$, we have*

$$
\left\|\frac{\boldsymbol{u}}{\|\boldsymbol{u}\|} - \frac{\boldsymbol{v}}{\|\boldsymbol{v}\|}\right\| \leq \frac{2\|\boldsymbol{u} - \boldsymbol{v}\|}{\max\{\|\boldsymbol{u}\|, \|\boldsymbol{v}\|\}}.
$$

**Lemma 17.** *Let $x_t = \arg\max_{x \in \mathcal{A}_t}\langle x, \hat{\boldsymbol{\theta}}_t\rangle$. We have*

$$
\max_{x \in \mathcal{A}_t}\langle x, \boldsymbol{\theta}_t\rangle - \langle x_t, \boldsymbol{\theta}_t\rangle \leq J\frac{\|\boldsymbol{\theta}_t - \hat{\boldsymbol{\theta}}_t\|^2}{\|\boldsymbol{\theta}_t\|}.
$$

*Proof.* For $\boldsymbol{\theta} \in \mathbb{R}^d$, we define $f_t(\boldsymbol{\theta}) = \max_{\boldsymbol{a} \in \mathcal{A}_t} \langle \boldsymbol{a}, \boldsymbol{\theta} \rangle$. Let $\boldsymbol{x}_t^* = \arg\max_{\boldsymbol{x} \in \mathcal{A}_t} \langle \boldsymbol{x}, \boldsymbol{\theta}_t \rangle$. Then we have

$$
\begin{aligned}
\max_{\boldsymbol{x} \in \mathcal{A}_t} \langle \boldsymbol{x}, \boldsymbol{\theta}_t \rangle - \langle \boldsymbol{x}_t, \boldsymbol{\theta}_t \rangle = \langle \boldsymbol{x}_t^* - \boldsymbol{x}_t, \boldsymbol{\theta}_t \rangle &= \langle \boldsymbol{x}_t^*, \boldsymbol{\theta}_t - \widehat{\boldsymbol{\theta}}_t \rangle + \langle \boldsymbol{x}_t^* - \boldsymbol{x}_t, \widehat{\boldsymbol{\theta}}_t \rangle + \langle \boldsymbol{x}_t, \widehat{\boldsymbol{\theta}}_t - \boldsymbol{\theta}_t \rangle \\
&\leq \langle \boldsymbol{x}_t^*, \boldsymbol{\theta}_t - \widehat{\boldsymbol{\theta}}_t \rangle + \langle \boldsymbol{x}_t, \widehat{\boldsymbol{\theta}}_t - \boldsymbol{\theta}_t \rangle = \langle \boldsymbol{x}_t^* - \boldsymbol{x}_t, \boldsymbol{\theta}_t - \widehat{\boldsymbol{\theta}}_t \rangle \\
&= \langle f_t(\boldsymbol{\theta}_t) - f_t(\widehat{\boldsymbol{\theta}}_t), \boldsymbol{\theta}_t - \widehat{\boldsymbol{\theta}}_t \rangle = \langle f_t(\frac{\boldsymbol{\theta}_t}{\|\boldsymbol{\theta}_t\|}) - f_t(\frac{\widehat{\boldsymbol{\theta}}_t}{\|\widehat{\boldsymbol{\theta}}_t\|}), \boldsymbol{\theta}_t - \widehat{\boldsymbol{\theta}}_t \rangle \\
&\leq \|f_t(\frac{\boldsymbol{\theta}_t}{\|\boldsymbol{\theta}_t\|}) - f_t(\frac{\widehat{\boldsymbol{\theta}}_t}{\|\widehat{\boldsymbol{\theta}}_t\|})\| \cdot \|\boldsymbol{\theta}_t - \widehat{\boldsymbol{\theta}}_t\| \leq J\|\frac{\boldsymbol{\theta}_t}{\|\boldsymbol{\theta}_t\|} - \frac{\hat{\boldsymbol{\theta}}_t}{\|\hat{\boldsymbol{\theta}}_t\|}\| \cdot \|\boldsymbol{\theta}_t - \hat{\boldsymbol{\theta}}_t\|
\end{aligned}
\tag{18}
$$

$$
\leq 2J \frac{\|\boldsymbol{\theta}_t - \hat{\boldsymbol{\theta}}_t\|^2}{\|\boldsymbol{\theta}_t\|},
\tag{19}
$$

where the first inequality in (18) uses Cauchy and (19) uses Lemma 16. $\qquad\square$

**Lemma 18.** *For each task* $t \in [T]$, *we have* $\mathbb{E}\|\widehat{\boldsymbol{\theta}}_t - \boldsymbol{\theta}_t\|^2 \lesssim \frac{k^2}{\lambda_1^2 N_2} + \|\widehat{\boldsymbol{B}}_\perp^\top \boldsymbol{B}\|^2$.

*Proof.* We define $\boldsymbol{\theta}'_t = \widehat{\boldsymbol{B}}\widehat{\boldsymbol{B}}^\top \boldsymbol{\theta}_t$. Note that $\boldsymbol{\theta}_t = \boldsymbol{\theta}'_t + \widehat{\boldsymbol{B}}_\perp \widehat{\boldsymbol{B}}_\perp^\top$. We have

$$
\begin{aligned}
\widehat{\boldsymbol{\theta}}_t - \boldsymbol{\theta}_t &= \widehat{\boldsymbol{B}}\widehat{\boldsymbol{w}}_t - (\widehat{\boldsymbol{B}}\widehat{\boldsymbol{B}}^\top \boldsymbol{\theta}_t + \widehat{\boldsymbol{B}}_\perp \widehat{\boldsymbol{B}}_\perp^\top)\boldsymbol{\theta}_t \\
&= \widehat{\boldsymbol{B}}(\widehat{\boldsymbol{w}}_t - \widehat{\boldsymbol{B}}^\top \boldsymbol{\theta}_t) - \widehat{\boldsymbol{B}}_\perp \widehat{\boldsymbol{B}}_\perp^\top \boldsymbol{B} w_t.
\end{aligned}
$$

Note that $\widehat{\boldsymbol{B}}$ is perpendicular to $\widehat{\boldsymbol{B}}_\perp$ and that $\|\widehat{\boldsymbol{B}}\| = \|\widehat{\boldsymbol{B}}_\perp\| = 1$. We have

$$
\|\widehat{\boldsymbol{\theta}}_t - \boldsymbol{\theta}_t\|^2 = \|\widehat{\boldsymbol{B}}(\widehat{\boldsymbol{w}}_t - \widehat{\boldsymbol{B}}^\top \boldsymbol{\theta}_t)\|^2 + \|\widehat{\boldsymbol{B}}_\perp \widehat{\boldsymbol{B}}_\perp^\top \boldsymbol{B} w_t\|^2 \leq \|\hat{w}_t - \widehat{\boldsymbol{B}}^\top \boldsymbol{\theta}_t\|^2 + \|\widehat{\boldsymbol{B}}_\perp^\top \boldsymbol{B}\|^2.
$$

Let $v_i = \sqrt{\lambda_0}\hat{b}_i$. The OLS estimator is given by

$$
\begin{aligned}
\hat{w}_t &= \left( \sum_{n=N_1+1}^{N_1+N_2+1} \widehat{\boldsymbol{B}}^\top v_i v_i^\top \widehat{\boldsymbol{B}} \right)^{-1} \sum_{n=N_1+1}^{N_1+N_2} \widehat{\boldsymbol{B}}^\top \boldsymbol{x}_{n,t} r_{n,t} \\
&= \left( \sum_{n=N_1+1}^{N_1+N_2+1} \widehat{\boldsymbol{B}}^\top \boldsymbol{x}_{n,t} \boldsymbol{x}_{n,t}^\top \widehat{\boldsymbol{B}} \right)^{-1} \sum_{n=N_1+1}^{N_1+N_2} \widehat{\boldsymbol{B}}^\top \boldsymbol{x}_{n,t} (\boldsymbol{x}_{n,t}^\top \widehat{\boldsymbol{B}} w'_t + \varepsilon_{n,t}) \\
&= w'_t + \left( \sum_{n=N_1+1}^{N_1+N_2+1} \widehat{\boldsymbol{B}}^\top \boldsymbol{x}_{n,t} \boldsymbol{x}_{n,t}^\top \widehat{\boldsymbol{B}} \right)^{-1} \sum_{n=N_1+1}^{N_1+N_2} \widehat{\boldsymbol{B}}^\top \boldsymbol{x}_{n,t} \varepsilon_{n,t}.
\end{aligned}
$$

Write $\boldsymbol{A} = \sum_{n=N_1+1}^{N_1+N_2+1} \widehat{\boldsymbol{B}}^\top \boldsymbol{x}_{n,t} \boldsymbol{x}_{n,t}^\top \widehat{\boldsymbol{B}}$.

$$
\begin{aligned}
\mathbb{E}\|\widehat{\boldsymbol{w}}_t - w'_t\|^2 &= \sum_{n=N_1+1}^{N_1+N_2} \boldsymbol{x}_{n,t}^\top \widehat{\boldsymbol{B}} \boldsymbol{A}^{-2} \widehat{\boldsymbol{B}}^\top \boldsymbol{x}_{n,t} \mathbb{E}\,\varepsilon_{n,t}^2 \\
&\leq \sum_{n=N_1+1}^{N_1+N_2} \boldsymbol{x}_{n,t}^\top \widehat{\boldsymbol{B}} \boldsymbol{A}^{-2} \widehat{\boldsymbol{B}}^\top \boldsymbol{x}_{n,t} \leq N_2\|\boldsymbol{A}^{-2}\| \leq N_2 (\frac{k}{\lambda_0 N_2})^2 = \frac{k^2}{\lambda_0^2 N_2}.
\end{aligned}
$$

Putting together, we have

$$
\mathbb{E}\|\widehat{\boldsymbol{\theta}}_t - \boldsymbol{\theta}_t\|^2 \leq \frac{k^2}{\lambda_0^2 N_2} + \|\widehat{\boldsymbol{B}}_\perp^\top \boldsymbol{B}\|^2. \qquad\square
$$

*Proof of Theorem 3.* Note that $J = O(1)$ under our assumptions, as indicated by Rusmevichientong & Tsitsiklis (2010). So we have

$$\mathbb{E}[R^{N,T}] \leq TN_1 + TN_2 + T(N - N_1 - N_2)J\frac{\mathbb{E}\|\hat{\boldsymbol{\theta}}_t - \boldsymbol{\theta}_t\|^2}{\|\boldsymbol{\theta}_t\|}$$

$$\leq TN_1 + TN_2 + TNJ\frac{\|\widehat{\boldsymbol{B}}_\perp^\top \boldsymbol{B}\|^2 + k^2/(\lambda_1^2 N_2)}{\omega}$$

$$\lesssim TN_1 + TN_2 + TN \cdot \frac{d^3 k^2}{N_1 T}\log^3(NT) + TN\frac{k^2}{N_2}$$

$$\leq TN_1 + \frac{Nd^3 k^2}{N_1}\log^3(NT) + TN_2 + TN\frac{k^2}{N_2}$$

$$\leq d^{1.5}k\sqrt{NT}\log^3(NT) + kT\sqrt{N}. \qquad \square$$

## E    PROOF OF THEOREM 4

In this appendix, we assume all action sets are spherical, i.e. $\mathcal{A}_{n,t} \equiv \mathbb{S}^{d-1}$ for $n \in [N], t \in [T]$. Note that these action sets meet Assumption 3.

For each task $t \in [T]$, we use $R^{N,(t)} = \sum_{n=1}^{N}[\max_{\boldsymbol{x} \in \mathcal{A}_{n,t}} \langle \boldsymbol{x}, \boldsymbol{\theta}_t \rangle - \mathbb{E}\langle \boldsymbol{x}_{n,t}, \boldsymbol{\theta}_t \rangle]$ to denote the regret incurred on task $t$.

**Lemma 19.** *Under the setting of Theorem 4, we have* $\inf_{\mathcal{A}}\sup_{\mathcal{I}} \mathbb{E}[R_{\mathcal{A},\mathcal{I}}^{N,T}] \geq CkT\sqrt{N}$, *where* $C = 0.0001$.

**Lemma 20.** *Under the setting of Theorem 4, we have* $\inf_{\mathcal{A}}\sup_{\mathcal{I}} \mathbb{E}[R_{\mathcal{A},\mathcal{I}}^{N,T}] \geq Cd\sqrt{kTN}$, *where* $C = 0.0001$.

*Proof of Theorem 4.* We combine Lemma 19 and Lemma 20. $\qquad \square$

The proofs in this appendix would largely follow the proofs in Appendix B, yet this appendix is much longer, because we need to construct instances that satisfies Assumption 3, Assumption 4, and Assumption 5.

Our proofs to the lemmas are based on the lower bounds for the (single-task) linearly parameterized bandit setting, which corresponds to the $T = 1$ case in our setting. For this single-task setting, we assume $k = d$ and $\boldsymbol{B} = \boldsymbol{I}_d$. (This setting need not meet Assumption 4 and Assumption 5). We write the regret as $R^N = R^{N,1}$ and call algorithms for the single-task setting as single-task algorithms.

We invoke the following lower bound for the single-task setting.

**Lemma 21.** *Assume $N \geq d^2$ and $d \geq 2$. Let $\mathcal{N}(\mu, \Sigma)$ be the multivariate normal distribution with mean $\mu$ and covariance matrix $\Sigma$. For any single-task algorithm, we have*

$$\mathbb{E}_{\boldsymbol{w} \sim \mathcal{N}(0, \mathbf{I}_d/d)}[R^N \cdot \mathbb{I}\{0.09 \leq \|\boldsymbol{w}\| \leq 3\}] \geq 0.006 d\sqrt{N}. \qquad (20)$$

The lemma can be proved by following the proof of Theorem 2.1 of Rusmevichientong & Tsitsiklis (2010). Next we use it to prove Lemma 19 and Lemma 20. The main idea to prove Lemma 19 is to note that we can treat our setting as $T$ independent $k$-dimensional linear bandits.

Let $\mu_d$ be the conditional probability measure whose density function is given by

$$f(x) = \begin{cases} \frac{g(x)}{\Pr_X[0.09 \leq \|X\| \leq 3]}, & 0.09 \leq \|x\| \leq 3, \\ 0, & \text{otherwise,} \end{cases}$$

where $X \sim \mathcal{N}(0, \boldsymbol{I}_d/d)$ is the multivariate gaussian vector and $g(x)$ is the probability density function of $X$. Then (20) implies

$$\mathbb{E}_{w \sim \mu_d}[R^N] \geq 0.006 d\sqrt{N}. \qquad (21)$$

*Proof of Lemma 19.* Without loss of generality, we assume $2k$ divides $T$. Suppose, for contradiction, that there is an algorithm $\mathcal{A}'$, such that for every instance $\mathcal{I}$, it incurs regret $\mathbb{E}[R^{N,T}_{\mathcal{A}',\mathcal{I}}] \leq CTk\sqrt{N}$. We replace the condition $\|w_t\| \leq 1$ by $\|w_t\| \leq 3$ in our setting. Note that $\mathcal{A}'$ implies an algorithm $\mathcal{A}$ that incurs regret $\mathbb{E}[R^{N,T}_{\mathcal{A},\mathcal{I}}] \leq 3CTk\sqrt{N}$.

We construct the following instances $\mathcal{I} = (\boldsymbol{B}, \boldsymbol{W})$, where $\boldsymbol{B} = \begin{pmatrix} \boldsymbol{I}_k & 0 \end{pmatrix}^{\top}$ and $\boldsymbol{W} = \begin{pmatrix} \boldsymbol{w}_1 & \cdots & \boldsymbol{w}_T \end{pmatrix}$ as follows. Let

$$\boldsymbol{w}_1 = \cdots \boldsymbol{w}_{T/2k} = \boldsymbol{e}_1, \boldsymbol{w}_{T/2k+1} = \cdots = \boldsymbol{w}_{T/k} = \boldsymbol{e}_2, \ldots, \boldsymbol{w}_{(k-1)T/2k+1} = \cdots = \boldsymbol{w}_{T/2} = \boldsymbol{e}_k,$$

where $\boldsymbol{e}_1, \ldots, \boldsymbol{e}_k \in \mathbb{R}^k$ is the standard basis. Let

$$\boldsymbol{w}_{T/2+1}, \ldots, \boldsymbol{w}_T \sim \mu_d$$

be i.i.d. drawn. Thanks to the first $\frac{T}{2}$ tasks, the instance $\mathcal{I}$ always satisfies Assumption 4. Note that $\|w_t\| \geq 0.09$, so the instance also satisfies Assumption 5. Then we have

$$\underset{\boldsymbol{w}_{T/2+1}, \ldots, \boldsymbol{w}_T \sim \mu_d}{\mathbb{E}} [R^{N,T}_{\mathcal{A},\mathcal{I}}] \leq 3CTk\sqrt{N}.$$

Thus

$$\sum_{t=T/2+1}^{T} \underset{\boldsymbol{w}_{T/2+1}, \ldots, \boldsymbol{w}_T \sim \mu_d}{\mathbb{E}} [R^{N,(t)}_{\mathcal{A},\mathcal{I}}] \leq \underset{\boldsymbol{w}_{T/2+1}, \ldots, \boldsymbol{w}_T \sim \mu_d}{\mathbb{E}} [R^{N,T}_{\mathcal{A},\mathcal{I}}] \leq 3CTk\sqrt{N}.$$

Therefore, we can find $t \in [T/2+1, T]$ such that

$$\underset{\boldsymbol{w}_{T/2+1}, \ldots, \boldsymbol{w}_T \sim \mu_d}{\mathbb{E}} [R^{N,(t)}_{\mathcal{A},\mathcal{I}}] \leq \frac{3CT}{T/2} k\sqrt{N} = 6Ck\sqrt{N}.$$

We note that the expectation operator $\mathbb{E}_{\boldsymbol{w}_{T/2+1}, \ldots, \boldsymbol{w}_T \sim \mu_d}$ is over all randomness on tasks $\tau \neq t$ and its parameter $w_\tau$. So there is a realization of the randomness on other tasks $\tau \neq t$ that satisfies

$$\underset{\boldsymbol{w}_t \sim \mu_d}{\mathbb{E}} [R^{N,(t)}_{\mathcal{A},\mathcal{I}} \mid \boldsymbol{w}_\tau, \varepsilon_\tau, \tau \neq t] \leq 6Ck\sqrt{N}.$$

Based on the realization $\boldsymbol{w}_\tau, \varepsilon_\tau$, we design a single-task algorithm $\mathcal{S}$, which plays task $t$ and simulates other tasks $\tau \neq t$ with $w_\tau, \varepsilon_\tau$. The algorithm achieves

$$\underset{\boldsymbol{w} \sim \mu_d}{\mathbb{E}} [R^{N}_{\mathcal{S},\mathcal{I}}] = \underset{\boldsymbol{w}_t \sim \mu_d}{\mathbb{E}} [R^{N,(t)}_{\mathcal{A},\mathcal{I}} \mid w_\tau, \varepsilon_\tau, \tau \neq t] \leq 6Ck\sqrt{N},$$

which contradicts to (21) because $C = 0.0001$. $\qquad\qquad\square$

*Proof of Lemma 20.* Without loss of generality, we assume that $d$ is even and that $2k$ divides $T$. Suppose, for contradiction, that there is an algorithm $\mathcal{A}'$, such that for every instance $\mathcal{I}$, it incurs regret $\mathbb{E}[R^{N,T}_{\mathcal{A}',\mathcal{I}}] \leq Cd\sqrt{kTN}$. We replace the condition $\|\boldsymbol{w}_t\| \leq 1$ by $\|\boldsymbol{w}_t\| \leq 3$ in our setting. Note that $\mathcal{A}'$ implies an algorithm $\mathcal{A}$ that incurs regret $\mathbb{E}[R^{N,T}_{\mathcal{A},\mathcal{I}}] \leq 3Cd\sqrt{kTN}$.

We prove the lemma separately, based on whether $k \geq \frac{d}{2}$ or not.

1. Consider $k \leq \frac{d}{2}$. We generate $k$ vectors $\psi_1, \ldots, \psi_k$ so that $\psi_i \sim \mu_{d-i+1}$. For every dimension $i$, we consider a map

$$c_i : \mathbb{R}^{d \times i} \to \mathbb{R}^{d \times (d-i+1)},$$
$$(x_1, \ldots, x_i) \mapsto (y_1, \ldots, y_{d-i+1}),$$

so that when $\{x_1, \ldots, x_i\}$ are orthogonal, the set $\{y_1, \ldots, y_{d-i+1}\}$ is the orthonormal basis of the orthogonal complement $\mathrm{span}\{x_1, \ldots, x_i\}^{\perp}$ of $\mathbb{R}^d$. Note that $c_i$ can be computed efficiently, e.g. by Gram-Schmidt process.

We define $\phi_1 = \psi_1$ and $\phi_i = c_{i-1}(\phi_1, \ldots, \phi_{i-1}) \cdot \psi_i$ for $i \geq 2$. We observe that $\{\phi_1, \ldots, \phi_k\}$ are orthogonal. Next we define our instance $\mathcal{I} = (\boldsymbol{B}, \boldsymbol{W})$, where $\boldsymbol{B} = (b_1 \quad \cdots \quad b_k)$ and $b_i = \frac{\phi_i}{\|\phi_i\|}$. For $\boldsymbol{W} = (w_1 \quad \cdots \quad w_T)$, we let

$$w_1 = \cdots = w_{T/k} = \|\phi_1\| \cdot \boldsymbol{e}_1,$$
$$w_{T/k+1} = \cdots = w_{2T/k} = \|\phi_2\| \cdot \boldsymbol{e}_2,$$
$$\vdots$$
$$w_{(k-1)T/k+1} = \cdots = w_T = \|\phi_k\| \cdot \boldsymbol{e}_k,$$

where $\{\boldsymbol{e}_1, \ldots, \boldsymbol{e}_k\} \subseteq \mathbb{R}^k$ is the standard basis. Note that the instance $\mathcal{I}$ always satisfies Assumption 4 and Assumption 5 by our choice of $\boldsymbol{w}_t$. Note that we have divided the tasks into $k$ groups, so that each group share the same vector $\boldsymbol{w}_t$. For $i = 1, \ldots, k$, we denote the regret of group $i$ as

$$R_{\mathcal{A},\mathcal{I}}^{N,((i))} = \sum_{t=(i-1)T/k+1}^{iT/k} R_{\mathcal{A},\mathcal{I}}^{N,(t)}.$$

For the algorithm $\mathcal{A}$, it incurs regret

$$\mathop{\mathbb{E}}_{\phi_1,\ldots,\phi_k} [R_{\mathcal{A},\mathcal{I}}^{N,((i))}] \leq 3Cd\sqrt{kNT}.$$

So there is a group $i \in [k]$, such that

$$\mathop{\mathbb{E}}_{\phi_1,\ldots,\phi_k} [R_{\mathcal{A},\mathcal{I}}^{N,((i))}] \leq \frac{1}{k} 3Cd\sqrt{kNT} = 3Cd\sqrt{\frac{NT}{k}}.$$

Similar to the proof of Lemma 19, we can fix the randomness for groups $j \neq i$ to obtain a realization, such that

$$\mathop{\mathbb{E}}_{\phi_i|\phi_{j(j\neq i)}} [R_{\mathcal{A},\mathcal{I}}^{N,((i))} \mid \phi_j, \varepsilon_j, j \neq i] \leq \frac{1}{k} 3Cd\sqrt{kNT} = 3Cd\sqrt{\frac{NT}{k}}.$$

Here we note that $\phi_i$ could depend on $\phi_\iota$ for $\iota \geq i + 1$. Now we let $\psi_i' \sim \mu_{d-k+1}$ and let

$$\phi_i' = A\psi_i', \qquad A = c_{k-1}(\phi_1, \ldots, \phi_{i-1}, \phi_{i+1}, \ldots, \phi_k). \tag{22}$$

Note that $\phi_i'$ and $\phi_i \mid \phi_{j(j\neq i)}$ are identical, so we have

$$\mathop{\mathbb{E}}_{\phi_i=\phi_i'} [R_{\mathcal{A},\mathcal{I}}^{N,((i))} \mid \phi_j, \varepsilon_j, j \neq i] \leq \frac{1}{k} 3Cd\sqrt{kNT} = 3Cd\sqrt{\frac{NT}{k}}. \tag{23}$$

We complete the proof by showing that (23) implies a single-task algorithm $\mathcal{S}$ that plays a $(d-k+1)$-dimensional linear bandit for $\frac{NT}{k}$ times. Let $w = \psi_i'$. Then $w$ is independently drawn from $\mu_{d-k+1}$. Next we design the algorithm $\mathcal{S}$, which runs $\mathcal{A}$ by playing the tasks $t \in \mathcal{T} = \{(i-1)T/k + 1, \ldots, iT/k\}$ and simulates other tasks $\tau \notin \mathcal{T}$. Note that playing the task $t \in \mathcal{T}$ is the same as playing the single-task bandit defined by $w$, because we have $\boldsymbol{\theta}_t = \boldsymbol{B}w_t = \phi_i = A\psi_i'$ and the matrix $A$ is known (as in (22)) after we fix the randomness on other tasks. Since $|\mathcal{T}| = \frac{T}{k}$ and $\mathcal{A}$ is played for $N$ times, $\mathcal{S}$ can play the single-task bandit specified by $w$ for $\frac{NT}{k}$ times. As a result, we have

$$\mathop{\mathbb{E}}_{\boldsymbol{w}\sim\mu_{d-k+1}} [R_{\mathcal{S},\mathcal{I}}^N(\mathcal{I})] = \mathop{\mathbb{E}}_{\phi_i=\phi_i'} [R_{\mathcal{A},\mathcal{I}}^{N,((i))} \mid \phi_j, \varepsilon_j, j \neq i]$$
$$\leq 3Cd\sqrt{\frac{NT}{k}}$$
$$\leq 6C(d-k+1)\sqrt{\frac{NT}{k}},$$

which contradicts to (21) because $C = 0.0001$.

2. Consider $k > \frac{d}{2}$. In this case, the lower bound in Lemma 19 becomes $\Omega(Tk\sqrt{N}) = \Omega(Td\sqrt{N})$. Since $T \geq k$, we have $d\sqrt{kTN} \lesssim Td\sqrt{N}$. Thus we conclude by Lemma 19. $\qquad\square$

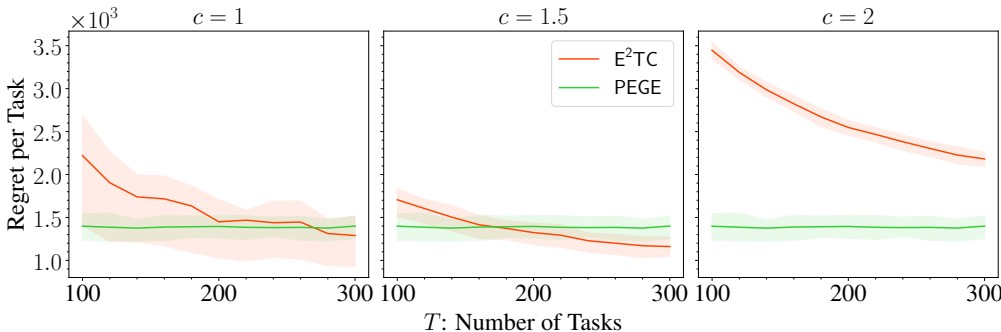

Figure 5: Comparisons of $\mathsf{E^2TC}$ with $\mathsf{PEGE}$ for $k = 3$.

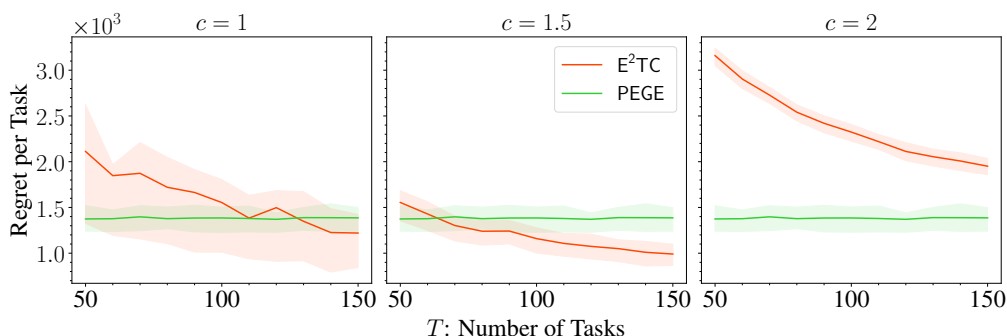

Figure 6: Comparisons of $\mathsf{E^2TC}$ with $\mathsf{PEGE}$ for $k = 2$.

## F    EXPERIMENTS FOR INFINITE-ARM SETTING

**Setup**    In all experiments, we set $d = 10, N = 10^4$ and the action $\mathcal{A}_t = \mathbb{S}^{d-1}$. The parameters are generated as follows. We consider $k = 2, 3$ in our experiments. The noise $\varepsilon_{n,t} \sim \mathcal{N}(0, 1)$ are i.i.d. Gaussian random variables. To verify our theoretical results, we consider a hyper-parameter $c \in \{0.5, 1, 1.5, 2\}$. For each $c$, we run $\mathsf{E^2TC}$ with $N_1 = d^c k \sqrt{\frac{N}{T}}$ and $N_2 = k\sqrt{N}$.

**Results and Discussions**    We present the simulation results in Figure 5 and Figure 6. We emphasize that the $y$-axis in our figures corresponds to the regret per task, which is defined as $\frac{\mathrm{Reg}_{N,T}}{T}$.

Our main observation is that only when the number of tasks $T$ is large and we choose the right scaling $N_1 = d^{1.5} k \sqrt{\frac{N}{T}}$, our method can outperform the $\mathsf{PEGE}$ algorithm. We discuss several implications of our results. First, representation learning does help, especially when there are many tasks available for us to learn the representation, as we see in all figures that the regret per task of $\mathsf{E^2TC}$ decreases as $T$ increases. Second, the help of representation learning is bounded. In the figures, we see that the curves of $\mathsf{E^2TC}$ bends to a horizontal line as $T$ increases, which suggests a lower bound on the regret per task. Meanwhile, we also proved an $\Omega(k\sqrt{N})$ lower bound on the regret per task in Theorem 4. Third, representation learning may have adverse effect without enough task. In our figures, this was established by noting that our algorithm cannot outperform $\mathsf{PEGE}$ when $T$ is small. This corresponds to the "negative transfer" phenomenon observed in previous work (Wang et al., 2019). Fourth, the correct hyper-parameter $c = 1.5$ is crucial. For hyper-parameter other than $c = 1.5$, the figures show that our algorithm would require much more tasks to outperform $\mathsf{PEGE}$. Lastly, by comparing the two figures, we notice that our algorithm has bigger advantage when $k$ is smaller, which corroborates the scaling with respect to $k$ in our regret upper bound. In contrast, $\mathsf{PEGE}$ does not benefit from a smaller $k$.

