# OpenReview forum: "Impact of Representation Learning in Linear Bandits"
_ICLR.cc/2021/Conference — ICLR 2021 Poster_

### Official Review · AnonReviewer2 · 2020-10-14
**The paper presents small step as an advancement towards a much needed theoretical support for a popular practice of representation learning in linear bandits. The authors support their findings by experimental evaluations.**

**Rating:** 7
**Confidence:** 3

**Review:**

- Pros.
   -  Learning the representation is crucial to model an efficient linear MAB. However, there is lot of room to include the characteristic of the learned representation into theoretical guarantee. This paper throws a light on this interesting problem.
   - The paper is well-written and the bounds look convincing. I have not gone through the detailed derivations (in the appendix), but the overall idea looks good.

- Cons.
   -  It would have been better if the paper could throw some light on other variants of representation learning.
   -  Linear bandit is quite popular news/ad recommendation systems. However, posing hand-writing recognition on MNIST data as linear bandit seems to be unnatural. There are DNN based approaches that solve the problem with a great accuracy. It will be interesting to see how does the algorithm perform on a real data set of news/ad recommendation.
   - Assumption 2 is a quite strong assumption to make.

I believe the exiting work: A Contextual-Bandit Approach to Personalized News Article Recommendation by Li et al (2010) deserves a citation in this paper.

---

> ### Author Response · Authors · 2020-11-15
> **Response to AnonReviewer2**
>
> We thank AnonReviewer2 for the positive view and valuable suggestions.  Please find our response to the comments below.
>
> 1. “It would have been better if the paper could throw some light on other variants of representation learning.”:
>
> We appreciate your advice and we have added a paragraph in the conclusion on representation learning with a general function class for the feature extractor.
>
> 2. “Linear bandit is quite popular news/ad recommendation systems…...”:
>
> Thanks for the suggestion! We will try to implement our algorithms on datasets of recommendation.
>
> 3. “Assumption 2 is a quite strong assumption to make.“
>
> We believe it is quite challenging to relax Assumption 2 to the adversarial contexts setting. One central challenge for the upper bound is that existing analyses for multi-task representation learning requires i.i.d. inputs even in the supervised learning setting. Another challenge is how to develop a confidence interval for an unseen input in the multi-task linear bandits setting. This confidence interval should utilize the common feature extractor and is tighter than the standard confidence interval for linear bandits, e.g. LinUCB. We have added this discussion in the conclusion (**Adversarial Contexts** paragraph).
>
> 4. Related work:
>
> We have cited the suggested related work in the introduction. Thanks for mentioning this work.

---

### Official Review · AnonReviewer3 · 2020-10-20
**Official Blind Review #3**

**Rating:** 6
**Confidence:** 4

**Review:**

*Summary*

This paper theoretically studies the benefits of representation learning in linear bandit problems. The key assumption is the existence of a common linear feature extractor. Two different setting are studied. In the finite-action setting, the authors provide the MLinGreedy algorithm that achieves matching upper and lower bounds (up to polylog factors). In the infinite-action setting, the authors provide the $E^2TC$ algorithm that can achieve lower regret than the naive method when the number of tasks is large. Experiments on both synthetic and real-world data are conducted, which confirm the theoretical results.

*Assessment*

Overall I'm leaning towards acceptance. Representation learning in sequential decision making problems is an important problem that many readers of ICLR will care about, and it is valuable to offer some theory insights into this problem. Besides, the paper is overall clearly written and enjoyable to read. Still I have some questions regarding the assumptions that the theoretical results are based on (see questions).

*Questions/Comments*
- The context vectors $x_{t,a}$ can be arbitrarily chosen in LinUCB (Chu et al. 2011). Is it possible to allow for adversarially chosen context in the finite-action setting?
- Intuitively, what is the reason that we do not need explicit exploration in MLinGreedy? I assume it has something to do with Lemma 3, which seems to rely on Assumption 2 ($\lambda_{\min}(\Sigma_t) \ge \Omega(1/d)$). Following the previous question, Assumption 2 looks rather strong; is there any good motivating example showing that Assumption 2 is likely to be true in practice?
- It would be better to give a concrete motivating example at the begining to give the readers a better idea of what $N, T, d, k$ look like order-wise.
- There's a gap between upper and lower bound in the infinite-action setting, but this is acceptable as a conference submission.

*Minor comments*
- Page 4, first line: $[w_1, \dots, w_T]$ should be in bold.

---

> ### Author Response · Authors · 2020-11-15
> **Response to AnonReviewer3**
>
> We thank AnonReviewer3 for the valuable comments and suggestions on our paper.  Please find our response to the comments below.
>
>
> 1. Is it possible to allow for adversarially chosen context in the finite-action setting?
>
> Thanks for asking. We’ve added a paragraph in the conclusion on this setting (**Adversarial Contexts** paragraph). This is indeed a challenging problem. One central challenge for the upper bound is that existing analyses for multi-task representation learning requires i.i.d. inputs even in the supervised learning setting. Another challenge is how to develop a confidence interval for an unseen input in the multi-task linear bandits setting. This confidence interval should utilize the common feature extractor and is tighter than the standard confidence interval for linear bandits, e.g. LinUCB.
>
>
> 2. Intuitively, what is the reason that we do not need explicit exploration in MLinGreedy? Is there any good motivating example showing that Assumption 2 is likely to be true in practice?
>
> The intuition is that Assumption 2 guarantees that by choosing the greedy action, the algorithm could explore other directions while exploiting the near-optimal direction $\hat{\theta}$. This observation has been used in some previous work, such as Han et al. (2020). Assumptions similar to our Assumption 2 also appeared in
> Bastani, Hamsa, Mohsen Bayati, and Khashayar Khosravi. "Mostly exploration-free algorithms for contextual bandits." Management Science (2020).
> In which the motivating example is in healthcare domains.
>
>
> 3. “It would be better to give a concrete motivating example at the beginning to give the readers a better idea of what $N, T, d, k$ look like order-wise.”
>
> Thanks for the suggestion! We’ve added a motivating example about the orders in the introduction.
>
>
> 4. “There's a gap between upper and lower bound in the infinite-action setting, but this is acceptable as a conference submission.”
>
> We’ve left it as an open problem for bridging this gap in Section 5.
>
> 5. Typos:
>
> Thanks for pointing out. We’ve fixed them accordingly.

---

### Official Review · AnonReviewer1 · 2020-10-28
**Good combination with representation learning and bandit**

**Rating:** 7
**Confidence:** 3

**Review:**

Summary:

This paper introduced the representation learning techniques into the linear bandits and showed that representation learning could improve the regret bound when multiple tasks shared a common low dimensional linear representation. The authors also proved a lower bound and extended the algorithm to the infinite-action setting. They also present experiments to validate their theoretical findings.

Pros:
(1) The idea of representation learning + linear bandits is very interesting.  Their results are also impressive which showed that combination will be better than naive algorithm.
(2) The paper is well written and easy to read. Their proof part looks good.

Cons:
 The assumptions (1,3,4) are based on the representation learning setting. Some of them seems very strong compared to the general linear bandits, e.g.,  assumption 4. Is this fair to compare your result with naive algorithm ? The naive algorithm could have better regret bound under the same setting. e.g., if the source tasks are diverse enough, the linear bandits will become much easier.

Minor Comments:
The author should use the better notations for bandit setting. It is strange that using T as the number of bandit tasks since T usually will be used as the time horizon of the experiments.

---

> ### Author Response · Authors · 2020-11-15
> **Response to AnonReviewer1**
>
>  We thank AnonReviewer1 for the positive review and the valuable comments on our paper. Below we address the cons you raised.
>
> 1. Some assumptions seem very strong compared to the general linear bandits. Is this fair to compare your result with naive algorithm?
>
> First, we would like to clarify that our Assumptions (3, 4) are only used when studying the infinite-action setting. They are not used for the finite-action setting. Second, for the infinite-action setting, the naive algorithm in our paper is to play $T$ linear bandits tasks independently. Since it plays independently, it cannot benefit from Assumption 4.
>
> 2. It is strange that using $T$ as the number of bandit tasks:
>
> We admit that $T$ is commonly used in bandits literature as the time horizon. However, $T$ is also commonly used in representation learning and multi-task learning literature as the number of tasks. So there's a conflict of notations between these two fields. To resolve it, we used $T$ as the number so as to be in line with the representation learning literature and we used $N$ to refer to the time horizon.

---

### Official Review · AnonReviewer4 · 2020-10-28
**An interesting problem but has several limitations**

**Rating:** 6
**Confidence:** 5

**Review:**

This paper studies the benefits of learning a low-rank feature extractor in multi-task linear bandits. Specifically, the paper studies the setting where an unknown common linear feature extractor $B \in R^{d \times k}$ maps the original $d$-dimensional contexts $x$ to a $k$-dimensional representation. Essentially, for multi-task linear bandit problem $r_t = \theta_t^T x$, this paper assumes the matrix of model parameters $\Theta \in R^{d\times T}$ is low-rank and be factorized as $\Theta = BW$ with rank k. The paper proposed algorithms to estimate both $B$ and $W$ in finite actions setting and infinite actions setting. In finite action setting, the proposed solution is a greedy algorithm while in infinite actions setting, the proposed solution is a explore-the-commit method. Theoretical result shows that the regret is $O(T\sqrt{kN} + \sqrt{dkTN})$ and matches the lower bound. Simulation result shows that the algorithm outperforms baselines running independent linear bandit algorithms.

Pros:
1) The problem studied in this paper is interesting and important. It is a well-known concern that linear bandits has a $O(d)$ regret for infinite actions and $O(\sqrt{d})$ regret for finite actions, which is too large for practical problem. Many recent studies aim to address this problem from varies perspectives, such as low-rank structure, sparsity, etc. The authors proposes to estimate a low-rank feature extractor for multi-task linear bandits to reduce the regret.

2) The paper is generally well-written and easy to follow (only a few confusing descriptions, mentioned below). The required assumptions are presented and discussed clearly.

3) The analysis of regret upper bound and lower bound are good contributions.


Cons:
1) The paper introduces a very strong assumption (Assumption 2) that the context features are sampled from Gaussian  (a stochastic context setting), and is a serious limitation of this paper. Most linear bandits research focus on the adversarial contexts setting, where the solution can also solve bandits with stochastic context but not the other way around. The gaussian feature assumption suggests that arms are equally informative and thus the classical exploration-exploitation dilemma does not exist.  It is not clearly discussed in the paper that why this paper must be limited to the stochastic feature setting. The paper heavily follows Theoretical results from [1] and both adversarial and stochastic contexts settings are discussed in [1]. What is the challenge to propose a solution for adversarial contexts, for example a linear UCB based solution?

2) It seems unclear on how to estimate both $B$ and $W$ and what is the estimation quality. According to the description the algorithm is doing an explicit matrix factorization. Since matrix factorization does not have a closed form solution, what is the guarantee of the estimation quality? In Lemma 2 the analysis requires the estimated $\hat B$ and $\hat W$ minimize the square loss so that the square loss with estimated $\hat B$ and $\hat W$ is smaller than the square loss with true $B$ and $W$(the first step of derivation)-- is this guaranteed to be achievable?

3) The authors argued that the problem of learning a low-rank feature extractor has not been studied in the bandit setting before, which seems incorrect: [2] studies a very similar problem that tries to estimate a hidden projection matrix for linear bandits with low-rank structure. In [2] the project matrix is estimated by PCA.  I strongly suggest the authors to compare this paper with [2]. One difference is that in [2] the low-rank structure is about the features (thus can be directly estimated from the features), while in this paper the low-rank structure is about the parameters $\Theta$. One potential advantages I can think of is this paper achieves regret in $O(\sqrt{k})$ while [2] has regret in $O(k)$ -- is it because of the assumption on features are sampled from gaussian?


Other questions:

1) What is the baseline naive algorithm in Figure 1? It seems to be linear regression + greedy strategy and I would suggest the authors to clarify it.

2) Is rank $k$ assumed to be an input to the algorithm? What if the algorithm does not know it or have a wrong knowledge of $k$?


==========================================================================================

While I still hold my concern on the i.i.d. assumption of the context as it is less interesting both practically and theoretically, the author response and the revised paper clearly resolve my other questions and concerns.  I am increasing my score to 6.


References:
[1] Han, Yanjun, Zhengqing Zhou, Zhengyuan Zhou, Jose Blanchet, Peter W. Glynn, and Yinyu Ye. "Sequential Batch Learning in Finite-Action Linear Contextual Bandits." arXiv preprint arXiv:2004.06321 (2020).
[2] Lale, Sahin, Kamyar Azizzadenesheli, Anima Anandkumar, and Babak Hassibi. "Stochastic linear bandits with hidden low rank structure." arXiv preprint arXiv:1901.09490 (2019).

---

> ### Author Response · Authors · 2020-11-15
> **Response to AnonReviewer4**
>
> We thank AnonReviewer4 for the valuable comments and suggestions. Please find our response to the comments below.
>
>
> 1. “The paper introduces a very strong assumption (Assumption 2) …”
>
> One central challenge for the upper bound is that existing analyses for multi-task representation learning requires i.i.d. inputs even in the supervised learning setting. Another challenge is how to develop a confidence interval for an unseen input in the multi-task linear bandits setting. This confidence interval should utilize the common feature extractor and is tighter than the standard confidence interval for linear bandits, e.g., LinUCB. We have added this discussion in the conclusion (**Adversarial Contexts** paragraph).
>
>
> 2. “What is the guarantee of the estimation quality of $\hat{B}, \hat{W}$?....”
>
> Note $(\hat{B}, \hat{W})$ is the minimizer of the empirical loss. We directly analyzed the statistical property of $(\hat{B}, \hat{W})$, and we measure the estimation quality in terms of the reconstruction of the linear predictors ($\Theta = BW$): $\lVert \hat{B}\hat{W} - BW\rVert_F^2$. See Lemma 4.
>
> 3. Connection with [2]:
>
> We appreciate you for pointing out the related paper. The main difference is that [2] assumed hidden feature extractor among actions, while we assumed the hidden feature extractor among tasks. Still, our Algorithm 2 has some similarities with [2], in that both used Davis-Kahan $\sin \Theta$ theorem to recover and exploit the low-rank structure.
>
> Regarding your question on the extra $O(\sqrt k)$ factor in the regret bound in [2], we believe it is mainly due to that [2] used a confidence ball similar to that constructed by [3], resulting in an extra factor depending on the dimension. For finite-action linear bandit, the analysis in [4] is more commonly used to obtain a tighter confidence interval.
>
>
> 4. What is the baseline naive algorithm in Figure 1?
>
> It's linear regression + greedy strategy as you correctly pointed out. We have updated the paper as well.
>
> 5. What if the algorithm does not know it or have a wrong knowledge of $k$?
>
> We assumed $k$ to be an input to the algorithm. If $k$ would be unknown, the algorithm could use an upper bound $k’$ of $k$, and the regret bound will degrade gracefully (replacing $k$ by $k’$ in the regret bound).
>
> Refs:
> [3] Abbasi-Yadkori, Yasin, Dávid Pál, and Csaba Szepesvári. "Improved algorithms for linear stochastic bandits." Advances in Neural Information Processing Systems. 2011.
> [4] Chu, Wei, et al. "Contextual bandits with linear payoff functions." Proceedings of the Fourteenth International Conference on Artificial Intelligence and Statistics. 2011.

---

### Official Review · AnonReviewer5 · 2020-11-05
**A theoretical study of the impact of a shared low-rank representation for multi-task linear bandits**

**Rating:** 7
**Confidence:** 4

**Review:**

This article provide a theoretical analysis of the impact of learning a shared low-rank representation in multi-task linear bandits.
Two types of linear bandits are considered: the finite actions and the continuous actions settings.
For the finite-actions setting the authors propose an algorithm called MLinGreedy and provide a theoretical analysis with a tight minimax regret bound (Theorem 1 & 2). For the continuous-actions setting they propose an "explore-then-explored-then-commit" algorithm called E2TC. They provide an upper-bound for the regret (Theorem 3), and a lower bound (Theorem 4). A gap remains between upper and lower regret bounds for this setting.
Theses bounds are roughly constituted of two parts: one part for the cost of learning the low-rank representation and one part which correspond to the cost of a linear bandit on a perfect low-rank representation.
In section 6 the authors provide two experiments for the finite actions setting. One experiment on a synthetic environment, another which is simulated-from "real-life" data. These experiments underline the advantage of learning a shared low-rank representation against a baseline where that tasks are considered independently. It does not expose situations where, according to the bounds, it should be at disadvantage.

If we omit the numerous typos that I listed at the end of this review, I found the paper clear and well written.
I did not have time to delve deeply into all the demonstrations, but the proof of Theorem 1 seems solid.
As explained on page 5 it relies on three lemma. Lemma 2 give guarantees on the low-rank approximation. It  is based on a Hoeffding concentration inequality combined with an epsilon-net argument. Lemma 3, borrowed from (Han et al., 2020), and Lemma 4 show that this estimate works well with the doubling schedule of MLinGreedy.

## Pro:
- The paper is well written
- The maths seem solid
- The results give a theoretical insight on the impact/benefit of representation learning in the specific case of linear bandits.

## Con:
- The multi-task linear bandit models rely on several assumptions and simplifications which obfuscate its realism. These assumptions are however justified thoroughly on page 3 and 4.
- I did not like the bias toward improvement of the introduction: for instance, according to the bound, if k is in Omega(d), the cost of learning the low-rank matrix is linear in d which results in a regret which is worse than the one of the "naive" approach for large dimensions. The authors should not be afraid to develop on the cases where the shared representation does not improve, it will not devaluate the significance of their work.

## Minor remarks & typos:
p1: I am not a fan of the "Provable benefits of" phrase in the title. This work is more about the impact of shared low-rank representation and when it can improves from the naive, but much simpler, independent-tasks approach. As mentioned on page 5 it requires T to be in Omega(k) to improve. "Impact of" would be more appropriate.
p1: "In this paper, we" -> "We"
p2: The \mathbold{x}_{n,t,a_{n,t}} notation is heavy and  not really needed. As mentioned on the bottom of page 3 one may interchangeably use \mathbold{a}_{n,t} and \mathbold{x}_{n,t,a_{n,t}} to refer the same action, so I would get rid of the heaviest notation.
p3: "line of work analyzed" -> "line of work that analyzed"
p3: "as our algorithm" -> "with our algorithm"
p3: "In this paper, we" -> "We"
p4: "In this assumption" -> "With this assumption"
p4: "for at each task" -> "for each task"
p4: "In each round" -> "At each round"
p4: "task are sample from" -> "task are sampled from"
p5: "up to an constant error" -> "up to a constant error"
p5: "we use an method-of" -> "we use a method-of"
p6:  "value decomposition of \hat{B}" -> "value decomposition of \hat{M}"
p7&8: the theoretical bounds should appear on Figure 2 and 4.
p8: Replacing the Time Horizon n with the number of tasks T as on Figure 2 would be more informative.

---

> ### Author Response · Authors · 2020-11-15
> **Response to AnonReviewer5**
>
> We thank AnonReviewer5 for the detailed comments and suggestions. Please find our response to the comments below.
>
> 1.  “The multi-task linear bandit models rely on several assumptions and simplifications ….”
>
> For theoretical developments, some assumptions and simplifications are necessary. We would like to note that these assumptions have appeared in previous bandits / representation learning theory literature. We have also added a paragraph in the conclusion on designing a robust algorithm which achieves our regret bounds when these assumptions are satisfied and gracefully reduces to the standard regret bounds of $T$ linear bandits when the assumptions do not hold.
>
> 2. “I did not like the bias toward improvement of the introduction.....”
>
> Thanks for the suggestion! This paper focuses on the positive impact of representation learning. We have added a discussion in the **robust algorithm** paragraph in conclusion.
>
> 3. Change of title:
>
> We have changed the title according to your suggestion. Thanks.
>
> 4. Typos:
>
> We thank you for pointing out the typos in our paper. We have fixed them accordingly. Besides, we will add the theoretical bounds to Figures 2 and 4, and replace $T$ with $N$ in Figure 4 in our next revision.

---

### Author Response · Authors · 2020-11-15
**Paper Revision**

We thank all the reviewers for the comments and suggestions! We have revised our paper accordingly. The main changes are shown in red fonts in the revised paper and are summarized as follows.

* Change the wording in the title.
* Section 1: Add a discussion on the relative orders of $d, k, N, T$.
* Section 2: Add extra related work on low-rank structure in linear bandits.
* Section 6: Add the description of the baseline (aka. naive) algorithm.
* Section 7: Add discussions on some future directions, including extending our results to the adversarial context setting and the generalized non-linear reward function setting.
* Fix the typos.

---

### Decision · Program_Chairs · 2021-01-07
**Final Decision**

**Decision:**

Accept (Poster)

**Comment:**

The paper studies the representation learning problem in the linear bandit setting, where each bandit "task" shares a common low-dimensional representation. The paper introduces a novel algorithm, it provides theoretical regret guarantees, and it illustrates the effectiveness of the proposed method in a number of experiments.

There is a general agreement among the reviewers about the relevance of the problem and the contribution of the paper. The authors properly addressed concerns about the novelty (e.g., comparison with linear bandit and low-rank structure) and about the underlying assumptions. Although some of them do seem relatively strong (and in some cases stronger than the state-of-the-art in bandit, such as the distribution on the contexts), it is indeed non trivial to understand whether such assumptions can be easily relaxed in the representation learning context.

The novelty of the algorithm is more on the specific problem and set of assumptions, but it mostly relies on known principles (e.g., using method-of-moment for estimating the underlying representation). In this sense, I see this paper more as a useful addition to the fast growing landscape of representation learning methods in online learning, rather than a breakthrough. Also, the structure of the algorithm seems very "theoretical" in nature, since the explore-than-commit approach is very rarely a good strategy in practice.

Another issue the authors clarified in their revised submission is the actual improvement obtained in the bounds depending on the parameters T, k, d, N. In this respect, I still would like to encourage the authors to further illustrate the regime where the bound is actually better than for the single-task approach. For instance, they could consider N fixed to a convenient value and produce a plot with x-axis T and y-axis the regret bound and report different curves for varying values of k and d. This would further clarify to the reader when representation learning can *provably* improve over plain single-task learning.

Overall, given the general support from the reviewers and the revised version of the paper, I consider this contribution is significant enough to propose acceptance. As mentioned above, I believe it will serve as a reference for developing further the literature in this domain.